# Long-term hematopoietic transfer of the anti-cancer and lifespan-extending capabilities of a genetically engineered blood system by transplantation of bone marrow mononuclear cells

Jing-Ping Wang[1,2†], Chun-Hao Hung[1,2*†], Yae-Huei Liou[2], Ching-Chen Liu[2], Kun-Hai Yeh[2], Keh-Yang Wang[1,2], Zheng-Sheng Lai[2], Biswanath Chatterjee[1,2], Tzu-Chi Hsu[1,2], Tung-Liang Lee[2,3,4], Yu-Chiau Shyu[2,5,6*], Pei-Wen Hsiao[7,8], Liuh-Yow Chen[2], Trees-Juen Chuang[9], Chen-Hsin Albert Yu[2], Nan-Shih Liao[2], C-K James Shen[1,2*]

[1]The Ph.D. Program in Medicine Neuroscience, Taipei Medical University, Taipei, Taiwan; [2]Institute of Molecular Biology, Academia Sinica, Taipei, Taiwan; [3]Chang Gung Memorial Hospital, Keelung, Taiwan; [4]Pro-Clintech Co. Ltd, Keelung, Taiwan; [5]Department of Nursing, Chang Gung University of Science and Technology, Taoyuan, Taiwan; [6]Community Medicine Research Center, Chang Gung Memorial Hospital, Keelung Branch, Keelung, Taiwan; [7]Agricultural Biotechnology Research Center, Academia Sinica, Taipei, Taiwan; [8]Graduate Institute of Life Sciences, National Defense Medical Center, Taipei, Taiwan; [9]Genomics Research Center, Academia Sinica, Taipei, Taiwan

*For correspondence:
libur777@gmail.com (C-HH);
yuchiaushyu@gmail.com (Y-CS);
ckshen@gate.sinica.edu.tw (C-KJS)

†These authors contributed equally to this work

**Abstract** A causal relationship exists among the aging process, organ decay and disfunction, and the occurrence of various diseases including cancer. A genetically engineered mouse model, termed *Klf1*^K74R/K74R or *Klf1*(K74R), carrying mutation on the well-conserved sumoylation site of the hematopoietic transcription factor KLF1/EKLF has been generated that possesses extended lifespan and healthy characteristics, including cancer resistance. We show that the healthy longevity characteristics of the *Klf1*(K74R) mice, as exemplified by their higher anti-cancer capability, are likely gender-, age-, and genetic background-independent. Significantly, the anti-cancer capability, in particular that against melanoma as well as hepatocellular carcinoma, and lifespan-extending property of *Klf1*(K74R) mice, could be transferred to wild-type mice via transplantation of their bone marrow mononuclear cells at a young age of the latter. Furthermore, NK(K74R) cells carry higher in vitro cancer cell-killing ability than wild-type NK cells. Targeted/global gene expression profiling analysis has identified changes in the expression of specific proteins, including the immune checkpoint factors PDCD and CD274, and cellular pathways in the leukocytes of the *Klf1*(K74R) that are in the directions of anti-cancer and/or anti-aging. This study demonstrates the feasibility of developing a transferable hematopoietic/blood system for long-term anti-cancer and, potentially, for anti-aging.

## eLife assessment

This **useful** article focuses on understanding how an Eklf mutation confers anti-cancer and longevity advantages in vivo. The data demonstrate that Eklf (K74R) imparts such advantages in a background- and age-independent manner in both female and male mice, and that the benefits are transferable

by bone marrow transplantation. Despite added data since a previous version, the article unfortunately remains **incomplete** as it is still unclear whether Eklf affects resistance to malignant progression/metastasis by modulating Pd1 or Pdl1 or by increasing NK cells. The authors provide evidence that supports in principle both mechanisms, and they do not resolve which mechanism is primarily involved.

## Introduction

Aging of animals, including humans, is accompanied by lifespan-dependent organ deterioration and the occurrence of chronic diseases such as cancer, diabetes, cardiovascular failure, and neurodegeneration (*Aman et al., 2021*; *López-Otín et al., 2013*). To extend healthspan and lifespan, various biomedical- and biotechnology-related strategies have been intensively developed and applied, including the therapy of different diseases such as cancer (*Bashor et al., 2022*; *Fontana and Partridge, 2015*; *Longo and Anderson, 2022*). The hematopoietic/blood system is an important biomedical target for anti-aging and anti-cancer research development. Multiple blood cell lineages arise from hematopoietic stem cells (HSCs), with the lymphoid lineage giving rise to T, B, and natural killer (NK) cell populations, whereas the myeloid lineage differentiates into megakaryocytes, erythrocytes, granulocytes, monocytes, and macrophages (*Mitchell et al., 2022*; *Pálovics et al., 2022*; *Seita and Weissman, 2010*). The genetic constituents and homeostasis of the hematopoietic system are regulated epigenetically and via environmental factors to maintain animal health (*Mitchell et al., 2022*; *Montazersaheb et al., 2022*).

EKLF, also named KLF1, is a Krüppel-like factor that is expressed in a range of blood cells, including erythrocytes, megakaryocytes, T cells, NK cells, as well as in various hematopoietic progenitors, including common myeloid progenitor , megakaryocyte erythroid progenitor (MEP), and granulocyte macrophage progenitor (*Frontelo et al., 2007*; *Nishizawa et al., 2016*; *Teruya et al., 2018*). The factor regulates erythropoiesis (*Perkins et al., 2016*) and the differentiation of MEP to megakaryocytes and erythrocytes (*Frontelo et al., 2007*; *Neuwirtova et al., 2013*) as well as of monocytes to macrophages (*Luo et al., 2004*). KLF1 is also expressed in HSC and regulates their differentiation (*Hung et al., 2020*). The factor can positively or negatively regulate transcription, including the adult β globin genes, through binding of its canonical zinc finger domain to the CACCC motif of the regulatory regions of a diverse array of genes (*Hung et al., 2020*; *Siatecka and Bieker, 2011*; *Pilon et al., 2011*; *Tallack et al., 2012*). In fact, the zinc finger of KLF1 could recognize/bind to genomic sites with the consensus sequence CCNCNCCC, and recruit different co-activator or co-repressor chromatin remodeling complexes (*Neuwirtova et al., 2013*; *Siatecka and Bieker, 2011*).

KLF1 could be sumoylated in vitro and in vivo, and sumoylation of the lysine at codon 74 of mouse KLF1 altered the transcriptional regulatory function (*Frontelo et al., 2007*) as well as nuclear import (*Shyu et al., 2014*) of the factor. Surprisingly, homozygosity of a single amino acid substitution, lysine(K) to arginine(R), at the sumoylation site of KLF1 results in the generation of a novel mouse model with healthy longevity. These mice, termed *Klf1*$^{K74R/K74R}$ or *Klf1*(K74R), exhibited extended healthspan and lifespan. In particular, the *Klf1*(K74R) mice showed delay of the age-dependent decline of physical performance, such as the motor function and spatial learning/memory capability, and deterioration of the structure/function of tissues, including the heart, liver, and kidney. Furthermore, the *Klf1*(K74R) mice appeared to have significantly higher anti-cancer capability than the WT mice (*Shyu et al., 2022*).

As described in the following, we have since characterized the high anti-cancer capability of the *Klf1*(K74R) mice with respect to its dependence on the age, gender, and genetic background. More importantly, we have demonstrated that the high anti-cancer ability of these genetically engineered mice could be transferred to wild-type mice (WT) through hematopoietic transplantation of the bone marrow mononuclear cells (BMMNCs). Furthermore, we show that the higher anti-cancer capability and extended lifespan of *Klf1*(K74R) mice are associated with changes in the global protein expression profile and specific aging-/cancer-associated cellular signaling pathways in their white blood cells (WBCs) or leukocytes.

## Results

### Characterization of the cancer resistance of *Klf1*(K74R) mice in relation to age, gender, and genetic background

The *Klf1*(K74R) mice appeared to be cancer resistant to carcinogenesis as manifested by their lower spontaneous cancer incidence (12.5%) in life than WT mice (75%). The *Klf1*(K74R) mutation also protected the mice from metastasis of melanoma cells and reduced melanoma growth in the subcutaneous cancer cell inoculation assay (*Shyu et al., 2022*). Here, we used the pulmonary melanoma foci assay, which has been commonly used for quick analysis of early metastasis (*Vaseghi et al., 2023*), to further characterize the higher cancer resistance of the *Klf1*(K74R) mice with respect to the effects of gender/age/genetic background of the mice and the requirement for the homozygosity of the K74R mutation. The cultured melanoma B16-F10 cells used in this assay are poorly immunogenic and highly aggressive, thus being useful for monitoring early metastasis in various immunotherapy studies (*Vaseghi et al., 2023*; *Díaz-García et al., 2018*).

It appeared that male as well as female *Klf1*(K74R) mice in the B6 genetic background had significantly fewer pulmonary melanoma foci than the corresponding WT mice (*Figure 1*). Because of this result, we used male mice for all of the studies described below. First, both young (2-month-old) and aged (24-month-old) *Klf1*(K74R) mice had higher anti-metastasis ability against the injected melanoma cells than WT mice of age-dependent groups (*Figure 1A and B*). Secondly, homozygosity of the K74R substitution was required for the higher cancer resistance of the *Klf1*(K74R) mice (*Figure 1—figure supplement 1*). Importantly, the *Klf1*(K74R) mice in the FVB background also exhibited a high cancer resistance than FVB WT mice by this assay (*Figure 1C*), suggesting that cancer resistance of *Klf1*(K74R) mice conferred by the homozygous K74R substitution was likely genetic background-independent. Finally, the higher anti-cancer capability of the *Klf1*(K74R) mice did not appear to depend on the arginine at codon 74 since *Klf1*(K74A) mice carrying K→A amino acid substitution at the K74 sumoylation site also exhibited higher anti-metastasis capability than WT mice in the pulmonary foci assay (*Figure 1D*).

The anti-cancer capability of *Klf1*(K74R) mice was not restricted to melanoma. As shown by the subcutaneous inoculation assay of Hepa1-6 cells (*Figure 1E*), *Klf1*(K74R) mice also carry a higher anti-cancer capability against hepatocellular carcinoma than the WT mice. It thus appears that the *Klf1*(K74R) mice are resistant to the carcinogenesis of a range of different cancers.

### Transfer of the anti-cancer capability and extended lifespan of *Klf1*(K74R) mice to WT mice via BMT

Since KLF1 is a hematopoietic transcription factor expressed not only in mature blood cells but also in HSCs and hematopoietic stem progenitor cells, this cancer resistance may be transferable by means of BMT. This possibility was tested by using male mice and a standard BMT protocol (*Imado et al., 2004*). BMMNCs were purified from the bone marrow of 2-month-old CD45.2 *Klf1*(K74R) or WT mice and injected into the tail vein of CD45.1 WT recipient mice. Blood replacement of recipient mice with 10 Gy γ-irradiation by that of the donor mice reached 90% in the seventh week (*Figure 2A and B*). After 2 wk, the recipient mice were injected with B16-F10 cells and then sacrificed a further 2 wk later to quantify pulmonary tumor foci. We found that WT mice transplanted with WT BMMNC had similarly high numbers of tumor foci relative to WT mice without BMT (*Figures 1A and 2C*). However, similar to *Klf1*(K74R) mice challenged with B16-F10 cells (*Figure 1B*), WT mice that received BMMNC from *Klf1*(K74R) mice presented significantly fewer tumor foci on their lungs (*Figure 2C*). Notably, BMT using 24-month-old donor mice gave similar results, although the effect is relatively small (*Figure 2—figure supplement 1A*).

In order to determine whether WT mice having more restricted blood replacement upon BMT from *Klf1*(K74R) mice still exhibited a higher anti-cancer capability, we also carried out BMT experiments with lower doses of γ-irradiation. BMT using two lower doses of γ-irradiation (2.5 Gy/5 Gy) still resulted in the transfer of cancer resistance from *Klf1*(K74R) to WT mice. Approximately 40% of recipient blood cells were substituted by donor cells upon BMT with 5 Gy γ-irradiation. Consequently, at that irradiation dosage, BMT from *Klf1*(K74R) mouse donors reduced the average number of pulmonary tumor foci in recipient WT mice to 5. On the other hand, only 20% blood replacement was achieved in the recipient mice with 2.5 Gy γ-irradiation (*Figure 2D*). However, the WT mice receiving BMT from

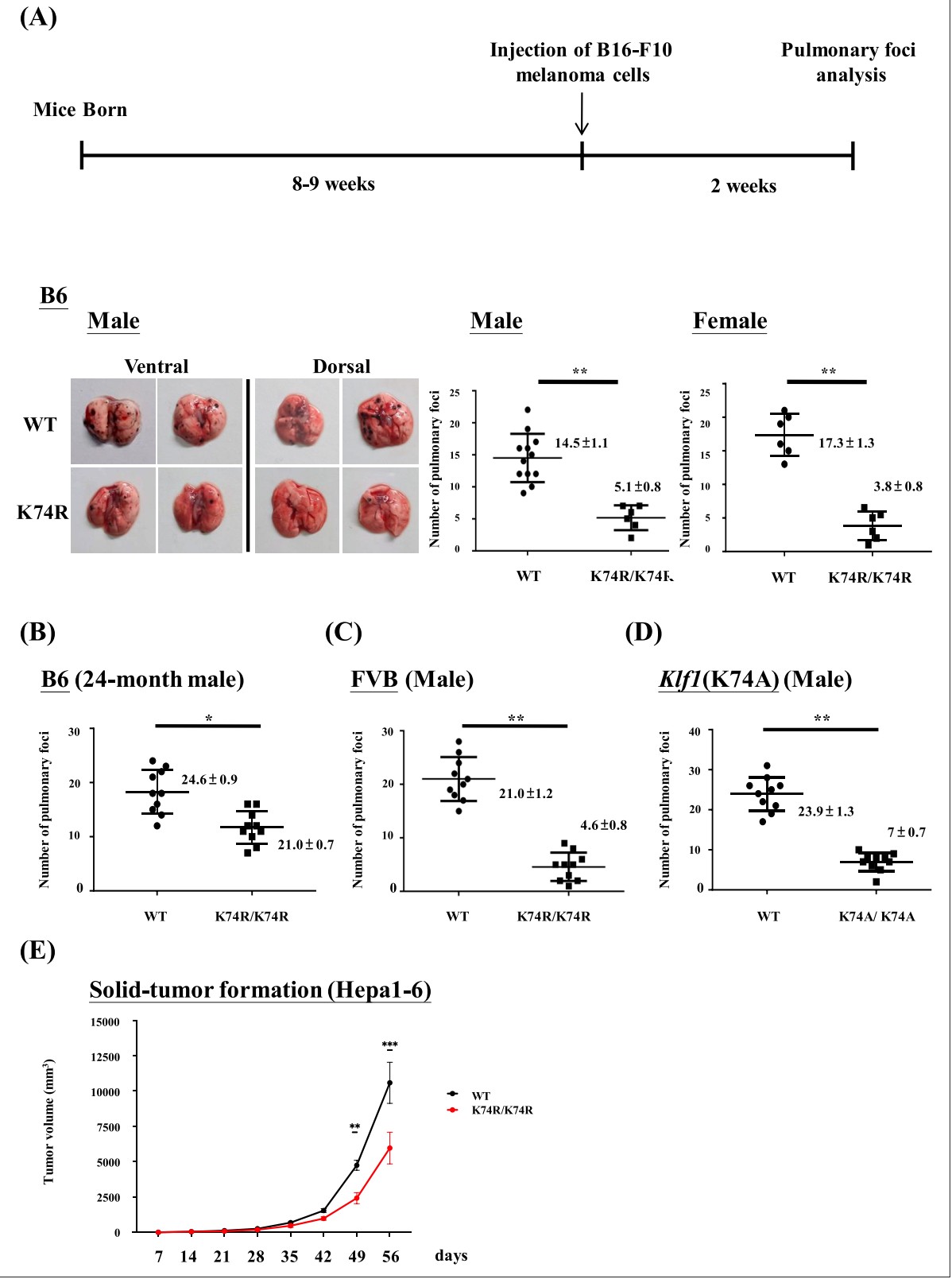

**Figure 1.** Anti-cancer capability of *Klf1*(K74R) mice as analyzed by the experimental melanoma metastasis assay. (**A**) Flow chart illustrating the strategy of the pulmonary tumor foci assay. Left panels: representative photographs of pulmonary metastatic foci on the lungs of wild-type (WT) and *Klf1*(K74R) male mice in the B6 background 2 wk after intravenous injection of B16-F10 cells ($10^5$ cells/ mouse). Statistical comparison of the numbers of pulmonary foci is shown in the two histograms on the right. N = 10 (male) and N = 7 (female), **p<0.01. Note that only the number of large pulmonary foci (>1 mm

*Figure 1 continued on next page*

*Figure 1 continued*

diameter) was scored. N > 6, **p<0.01. (**B**) Pulmonary tumor foci assay of 24-month-old WT and *Klf1*(K74R) male mice. Statistical comparison is shown in the two histograms. N = 10 (male), *p<0.05. (**C**) Pulmonary tumor foci assay of male mice in the FVB background. Statistical comparison is shown in the histograph on the right. N = 10, **p<0.01. (**D**) Pulmonary tumor foci assay of *Klf1*(K74A) male mice. Statistical comparison of the 3-month-old WT and *Klf1*(K74A) mice numbers of pulmonary foci is shown in the two histograms. N = 10 (male), **p<0.01. (**E**) The *Klf1*(K74R) mice and WT mice were subcutaneously injected with Hepa1-6 cells to form tumors. The tumor volumes were measured once a week using the formula: length × weight × height × π/6. The curves of tumor growth started to show difference between the *Klf1*(K74R) and WT mice at 49 d after Hepa1-6 cell injection. N > 6, *p<0.05, **p<0.01, and ***p<0.001.

The online version of this article includes the following figure supplement(s) for figure 1:

**Figure supplement 1.** Cancer resistance of *Klf1*(K74R) homozygous and heterozygous mice.

*Klf1*(K74R) mice again developed less number (~10/mouse) of pulmonary tumor foci than those WT mice receiving BMT from the WT mice (***Figure 2D***). Thus, even at a low level of 20% blood replacement, BMT still enabled effective transfer of cancer resistance from *Klf1*(K74R) mice to WT mice.

In addition, we also attempted to transfer the extended lifespan characteristics of the *Klf1*(K74R) mice to WT mice by BMT. Significantly, the median lifespan of WT mice receiving BMT from *Klf1*(K74R) mice was 5 mo longer than that of WT mice receiving BMT from WT mice (***Figure 2—figure supplement 1B***). Thus, the longer lifespan characteristics of the *Klf1*(K74R) mice were also transferable via BMT.

## Inhibition of tumor growth by transplanted BMMNC from *Klf1*(K74R) mice

Our experiments indicated that *Klf1*(K74R) bone marrow carried the anti-metastasis capability that prevented melanoma cell colonization in the lungs of recipient mice (***Figures 1 and 2***). To determine whether *Klf1*(K74R) BMT could inhibit tumor growth, we examined the effects of BMT on growth of tumors with B16-F10-luc cells. As outlined in ***Figure 3A***, 10 d after injection of cancerous cells, the formation of bioluminescent signals in the recipient mice was confirmed by the observation of in vivo bioluminescence. The following day, we transplanted the recipient mice with BMMNC from WT or *Klf1*(K74R) mice and then measured the intensities of bioluminescence signals from tumors 7 and 14 d later. As shown, tumor growth in mice subjected to BMT from *Klf1*(K74R) mice was significantly slower relative to those receiving BMMNC from WT mice (***Figure 3B and C***). Thus, *Klf1*(K74R) BMMNC indeed can inhibit the growth of tumors more effectively than WT BMMNC.

## Differential expression of specific immune-, aging-, and/or cancer-associated biomolecules in the blood of *Klf1*(K74R) mice

In general, the cell populations in peripheral blood (PB) were not affected much by the K74R substitution, as shown by CBC analysis (***Shyu et al., 2022***), although flow cytometry analysis of PB from WT and *Klf1*(K74R) mice of different ages showed that the population of NK(K74R) cells in 24-month-old *Klf1*(K74R) mice was higher than the WTNK cells in 24-month-old WT mice (***Figure 4—figure supplement 1***). In parallel, the K74R substitution did not affect much the expression of *Klf1* gene in the different types of hematopoietic cells. As shown in ***Figure 4—figure supplement 2***, the K74R substitution did not alter the expression levels of KLF1 protein in leukocytes from PB (***Figure 4—figure supplement 2A***, left panels), although the *Klf1* mRNA level of *Klf1* (K74R) leukocytes was 10–20% lower than that of WT leukocytes (***Figure 4—figure supplement 2A***, right panel). Previous studies and databases have shown that *Klf1* mRNA and KLF1 protein are expressed in the erythroid lineage (***Frontelo et al., 2007***) and megakaryocytes (***Frontelo et al., 2007***; ***Neuwirtova et al., 2013***), and the *Klf1* mRNA is expressed in HSC (***Hung et al., 2020***) (bio-GPS database), NK cells (bio-GPS database), naïve T cells (***Teruya et al., 2018***), and Treg cells (***Teruya et al., 2018***). By western blotting (WB) analysis (***Figure 4—figure supplement 2B***) and RT-qPCR analysis (***Figure 4—figure supplement 2C***), we found that the *Klf1* gene is expressed in B220[+] B cells, CD3[+] T cells, and NK cells of both WT and *Klf1*(K74R) mice, but the levels are much lower than that of the erythroid MEL cells.

We first used RT-qPCR to analyze the levels in PB cells of mRNAs encoding the immune checkpoint genes (ICGs) PDCD/CD274 (***Iwai et al., 2017***) in view of the cancer resistance of *Klf1*(K74R) mice (***Figure 4***) as well as increased levels of PDCD and CD274 in aged or tumorigenic mice (***Jiang et al.,***

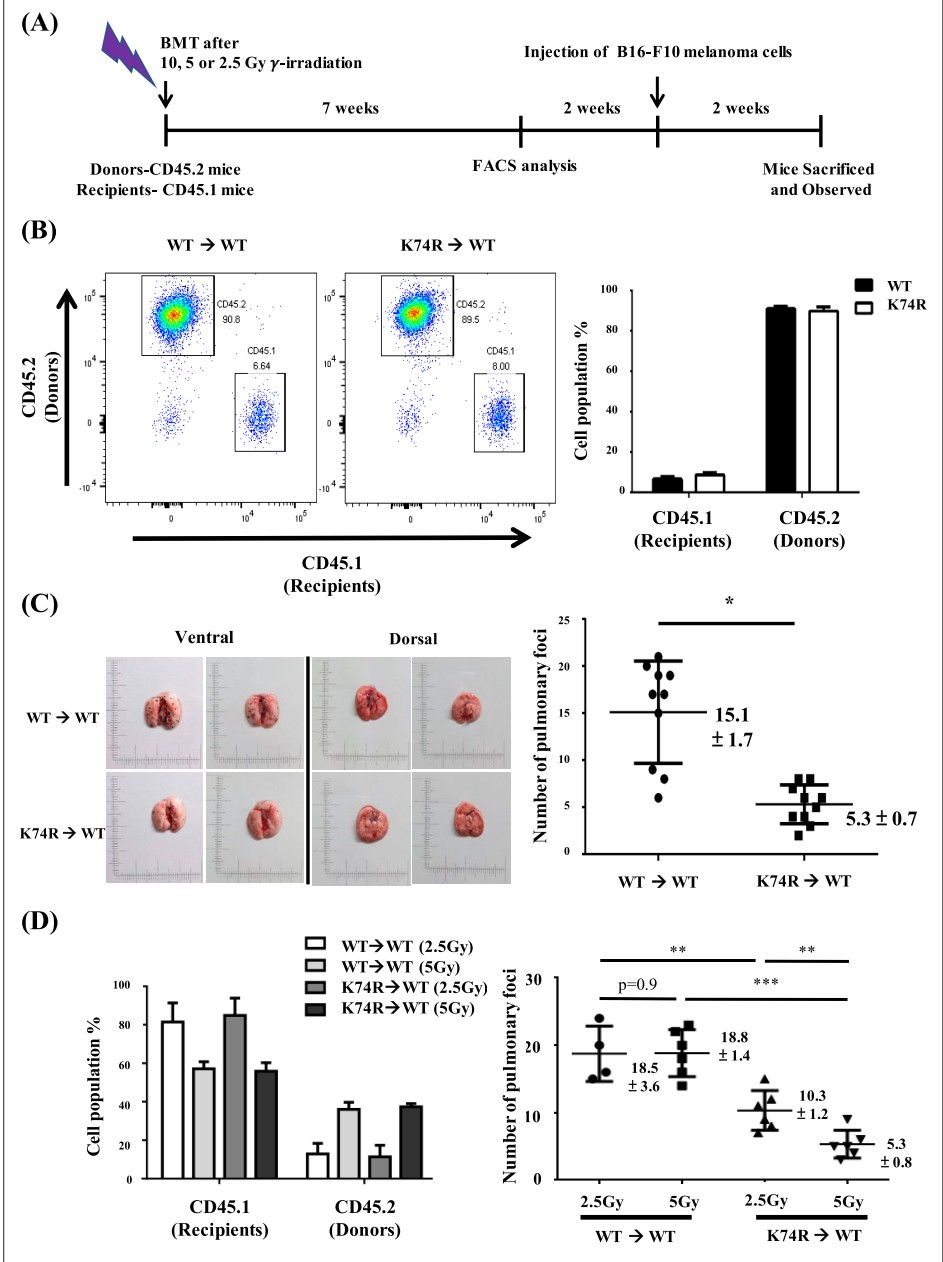

**Figure 2.** Transfer of cancer resistance of *Klf1*(K74R) mice to wild-type (WT) mice by bone marrow transplantation (BMT). (**A**) Flow chart illustrating the experimental strategy. (**B**) Fluorescence-activated cell sorting (FACS) analysis of the efficiency of BMT by using 10 Gy γ-irradiation. The percentages of CD45.1/CD45.2 cells in the peripheral blood (PB) of the recipient male mice were analyzed by flow cytometry, with the representative FACS charts shown on the left and the statistical histobar diagram on the right. (**C**) Transfer of the anti-metastasis capability of 8-week-old *Klf1*(K74R) male mice to age-equivalent WT male mice by BMT by using 10 Gy γ-irradiation. Left panels: representative photographs of lungs with pulmonary metastatic foci in the recipient WT (CD45.1) mice after BMT from WT (CD45.2) or *Klf1*(K74R) (CD45.2) donor mice and challenged with B16-F10 cells. Statistical analysis of the number of pulmonary B16-F10 metastatic foci on the lungs is shown in the right histogram. n = 10, *p<0.05. (**D**) Transplantation of 8-week-old male WT (CD45.1) mice with BMMNC from age-equivalent WT (CD45.2) male mice or from *Klf1*(K74R) (CD45.2) male mice by using the γ-irradiation dosage 2.5 Gy or 5 Gy. The histobar diagram comparing the percentages of CD45.1 and CD45.2 PB cells of the recipient WT mice after BMT is shown on the left. The statistical analysis of the average number of pulmonary foci on the lungs of recipient WT mice after BMT and injected with the B16-F10 cells is shown in the right histogram, N = 6. **p<0.01, ***p<0.001.

The online version of this article includes the following figure supplement(s) for figure 2:

*Figure 2 continued on next page*

*Figure 2 continued*

**Figure supplement 1.** Transfer of cancer resistance of aging Klf1(K74R) mice to young wild-type (WT) mice by bone marrow transplantation (BMT), and lifespan prolonging by young mice bone marrow transplantation.

*2019*). As shown in *Figure 4*, the mRNA levels of *Pdcd* and *Cd274* in the PB, B cells, and T cells of WT mice were both increased during aging. In contrast, the mRNA levels and protein levels of these two genes were lower in 3-month-old *Klf1*(K74R) mice than the age-matched WT mice, and they remained low during aging of the *Klf1*(K74R) mice. Significantly, RNAi knockdown of *Klf1* expression reduced

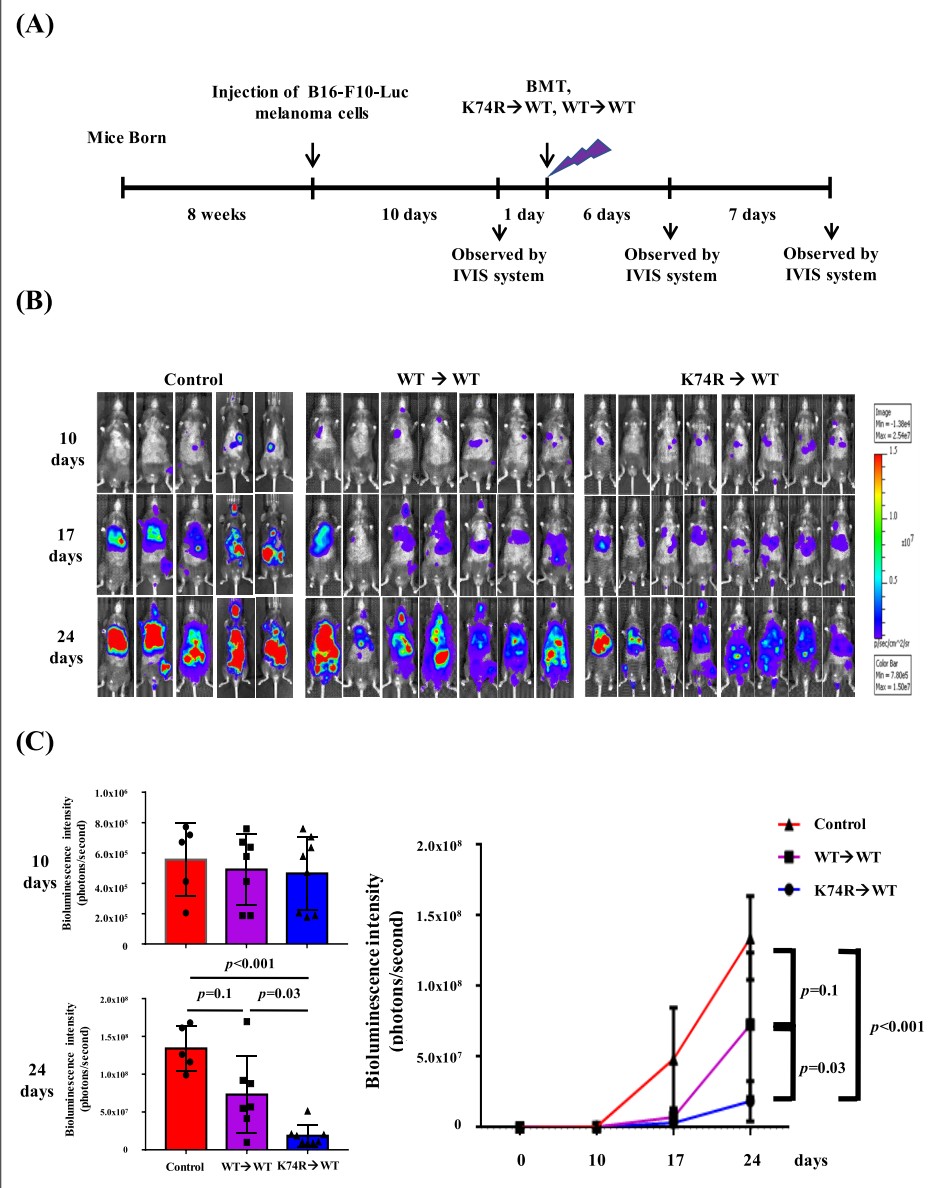

**Figure 3.** Inhibition of tumor growth in wild-type (WT) mice by bone marrow transplantation (BMT) from *Klf1*(K74R) mice. (**A**) A flow chart of the experiments. Luciferase-positive B16-F10 cells were injected into the tail vein of 8-week-old WT male mice (day 0). The mice were then transplanted with bone marrow mononuclear cells (BMMNCs) from WT or *Klf1*(K74R) male mice on day 11 after the luciferase-positive B16-F10 cell injection. In vivo imaging system (IVIS) was used to follow the tumor growth in mice on days 0, 10, 17, and 24, respectively. (**B**) Representative images of bioluminescence reflecting the luciferase activity from melanoma cancer cells in mice. The color bar indicates the scale of the bioluminescence intensity. (**C**) Statistical analysis of the intensities of bioluminescence in the cancer-bearing mice (WT→WT, purple, N = 7; *Klf1*(K74R)→WT, blue, N = 8; control [no BMT], red, N = 3).

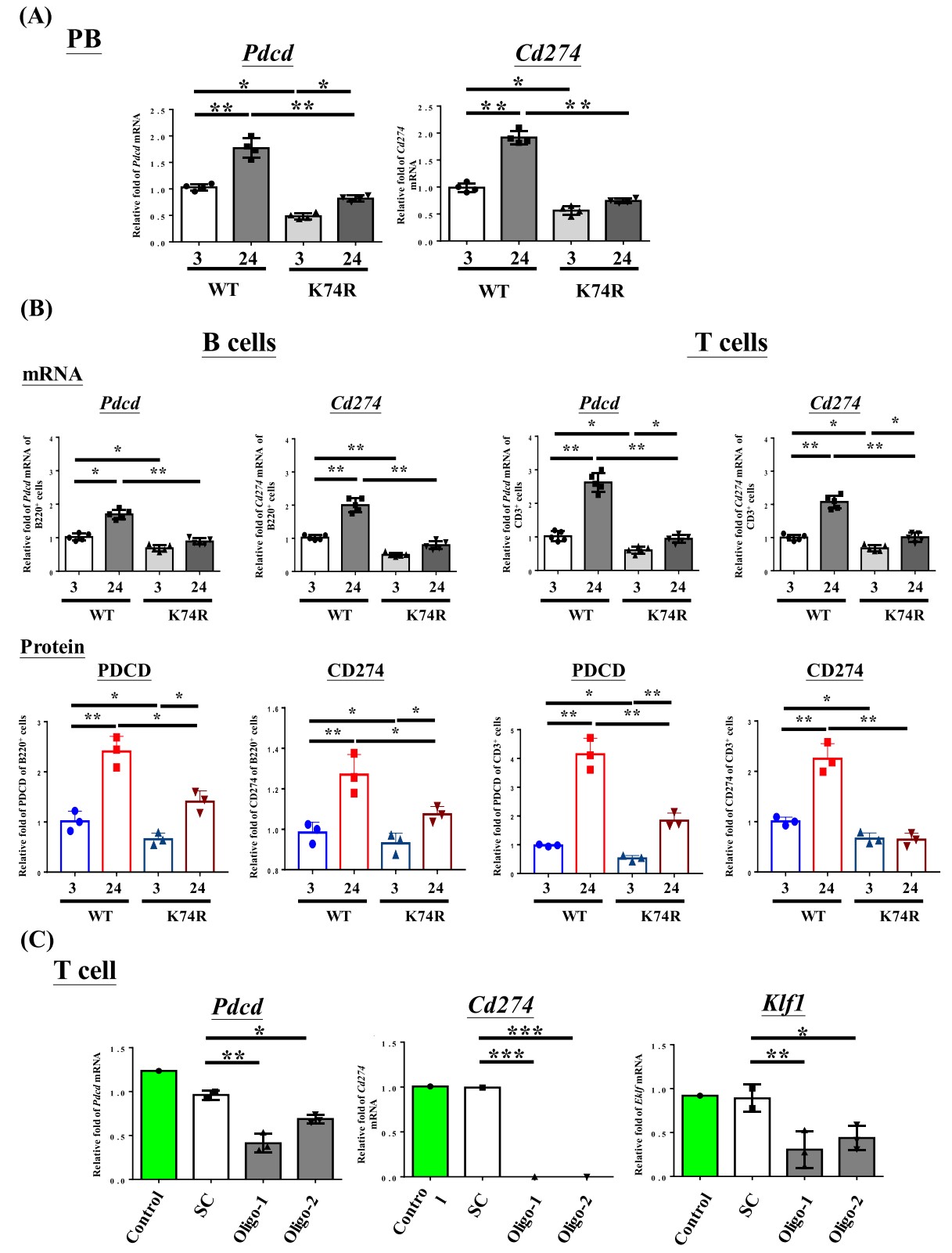

**Figure 4.** Decrease in *Pdcd* and *Cd274* expression in blood cells of *Klf1*(K74R) mice. (**A**) Levels of *Pdcd* and *Cd274* mRNAs in the peripheral blood (PB) of wild-type (WT) and *Klf1*(K74R) male mice at the ages of 3 mo and 24 mo, respectively, as analyzed by RT-qPCR. Note the relatively low levels of *Pdcd* and *Cd274* mRNAs in the *Klf1*(K74R) mice at both ages in comparison to the WT mice. (**B**) Upper panels: comparison of the mRNA levels of *Pdcd* and *Cd274* of CD3⁺ T cells and B220⁺ B cells isolated from the PB of 8-week-old WT and *Klf1*(K74R) male mice. N = 5. *p<0.05; **p<0.01. Lower

*Figure 4 continued on next page*

Figure 4 continued

panels: comparison of the protein levels of PDCD and CD274, as analyzed by flow cytometry, of CD3$^+$ T cells and B220$^+$ B cells from 8-week-old WT and *Klf1*(K74R) male mice. N = 3. *p<0.05; **p<0.01. (**C**) Comparison of the levels of *Pdcd, Cd274,* and *Klf1* mRNAs, as analyzed by RT-qPCR, in CD3$^+$ T cells, which were isolated from splenocytes, without or with RNAi knockdown of *Klf1* mRNA. Two oligos (oligo-1 and oligo-2) were used to knock down *Klf1* mRNA by ~60–70%, which resulted in the reduction of *Pdcd* mRNA level by 30–60% and nearly complete depletion of *Cd274* mRNA. Control, T cells transfected with GFP-plasmid; SC, T cells transfected with scrambled oligos. N > 3. *p<0.05; **p<0.01; ***p<0.001.

The online version of this article includes the following source data and figure supplement(s) for figure 4:

**Figure supplement 1.** Comparison of hematopoietic cell populations between wild-type (WT) and *Klf1*(K74R) mice.

**Figure supplement 2.** Comparative analysis of *Klf1* gene expression in hematopoietic cells of wild-type (WT) and *Klf1*(K74R) mice.

**Figure supplement 2—source data 1.** Original file for the western blot analysis in *Figure 4—figure supplement 2A* (anti-KLF1, anti-β-actin).

**Figure supplement 2—source data 2.** Original file for the western blot analysis in *Figure 4—figure supplement 2B* (anti-KLF1, anti-β-actin).

**Figure supplement 3.** ChIP-qPCR analysis of KLF1-binding on the *Pdcd* promoter of CD3$^+$ T cells from wild-type (WT) and *Klf1*(K74R) mice.

**Figure supplement 4.** Comparison of the serum levels of different cytokines in wild-type (WT) and *Klf1*(K74R) mice.

*Pdcd* and *Cd274* mRNA levels in splenic CD3$^+$ T cells (*Figure 4C*). In an interesting correlation, overexpression of KLF1 would increase the expression of *Cd274* in mouse CD4$^+$ T cells (*Teruya et al., 2018*). These data together indicate that KLF1 positively regulates the transcription of the *Pdcd* and *Cd274* genes, directly or indirectly, and likely the transcriptomes of a diverse range of hematopoietic cells. Since the levels of KLF1 protein in CD3$^+$ T (K74R) cells and WT CD3$^+$ T cells were similar (*Figure 4—figure supplement 2B*), change of the expression of *Pdcd* and *Cd274* genes (*Figure 4*) was most likely caused by the loss of the sumoylation of KLF1(K74R), which would alter its transactivation capability (*Siatecka and Bieker, 2011*). Interestingly, *Pdcd* appeared to be a direct regulatory target of KLF1. As shown by ChIP-qPCR analysis, KLF1 was bound at the CACCC box (Box3) at –103 of the *Pdcd* promoter in CD3$^+$ T cells from both WT and *Klf1*(K74R) mice (*Figure 4—figure supplement 3*). The details of the role of sumoylation in the transcriptional activation of *Pdcd* by KLF1 await future investigation. As expected, lower levels of *Pdcd* and *Pdcd1* expression were also observed in the PB of mice receiving BMT from *Klf1*(K74R) mice (*Figure 4—figure supplement 2D*). These findings together suggest that the low tumorigenesis rate of *Klf1*(K74R) mice arises in part from the low expression of the two ICGs, *Pdcd* and *Cd274*, as a result of K74R substitution in the KLF1 protein.

We also examined, by bead-based multiplex assay (*Aira et al., 2019*; *Menees et al., 2021*), the expression patterns of several aging- and/or cancer-associated cytokines. As shown in *Figure 4—figure supplement 4*, there was no significant difference in the serum levels of IL-1β, IL-2, IL-10, IL-12p70, INF-γ, or TNF-α between WT and *Klf1*(K74R) mice at 24-month-old. In contrast, the level of IL-4, an anti-inflammatory cytokine (*Ul-Haq et al., 2016*) beneficial to the hippocampus of aging mice (*Boccardi et al., 2018*), in 24-month-old *Klf1*(K74R) mice was three- to fourfold higher than the 24-month-old WT mice. On the other hand, the level of IL-6, a key factor in chronic inflammatory diseases, autoimmunity, cytokine storm, and cancer (*Menees et al., 2021*; *Rea et al., 2018*), increased only moderately during aging of the *Klf1*(K74R) mice (*Figure 4—figure supplement 4*). Thus, similar to PDCD and CD274, the altered expression of some of the cytokines in the blood likely also contributes to the anti-aging and/or anti-cancer characteristics of the *Klf1*(K74R) mice.

## Comparative proteomics analysis of the leukocytes of *Klf1*(K74R) mice and WT mice

We proceeded to examine age-dependent cell-intrinsic changes in the proteomes of the leukocytes from the WT and *Klf1*(K74R) mice in two different age groups. A total of 259 and 306 differentially expressed proteins (DEPs) were identified between the two age groups for the WT and *Klf1*(K74R) mice, respectively (*Figure 5A*). To understand the correlations of these proteins with aging and cancer, we performed the Gene Set Enrichment Analysis (GSEA) and found that the age-dependent DEPs changed in the concordant direction in the WT and *Klf1*(K74R) mice were enriched for several known aging-related pathways, for example, IL-6-JAK-STAT3 signaling, DNA repair, etc. (*Fabian et al., 2021*; *Figure 5B*). Meanwhile, the age-dependent DEPs changed in the reverse directions in the WT and *Klf1*(K74R) mice were enriched for nine other aging-related pathways (*Figure 5C*).

We further performed DEP analyses between WT and *Klf1*(K74R) mice and identified strain-dependent DEPs in the two age groups. As shown in *Figure 5D*, only 7 DEPs were identified in the

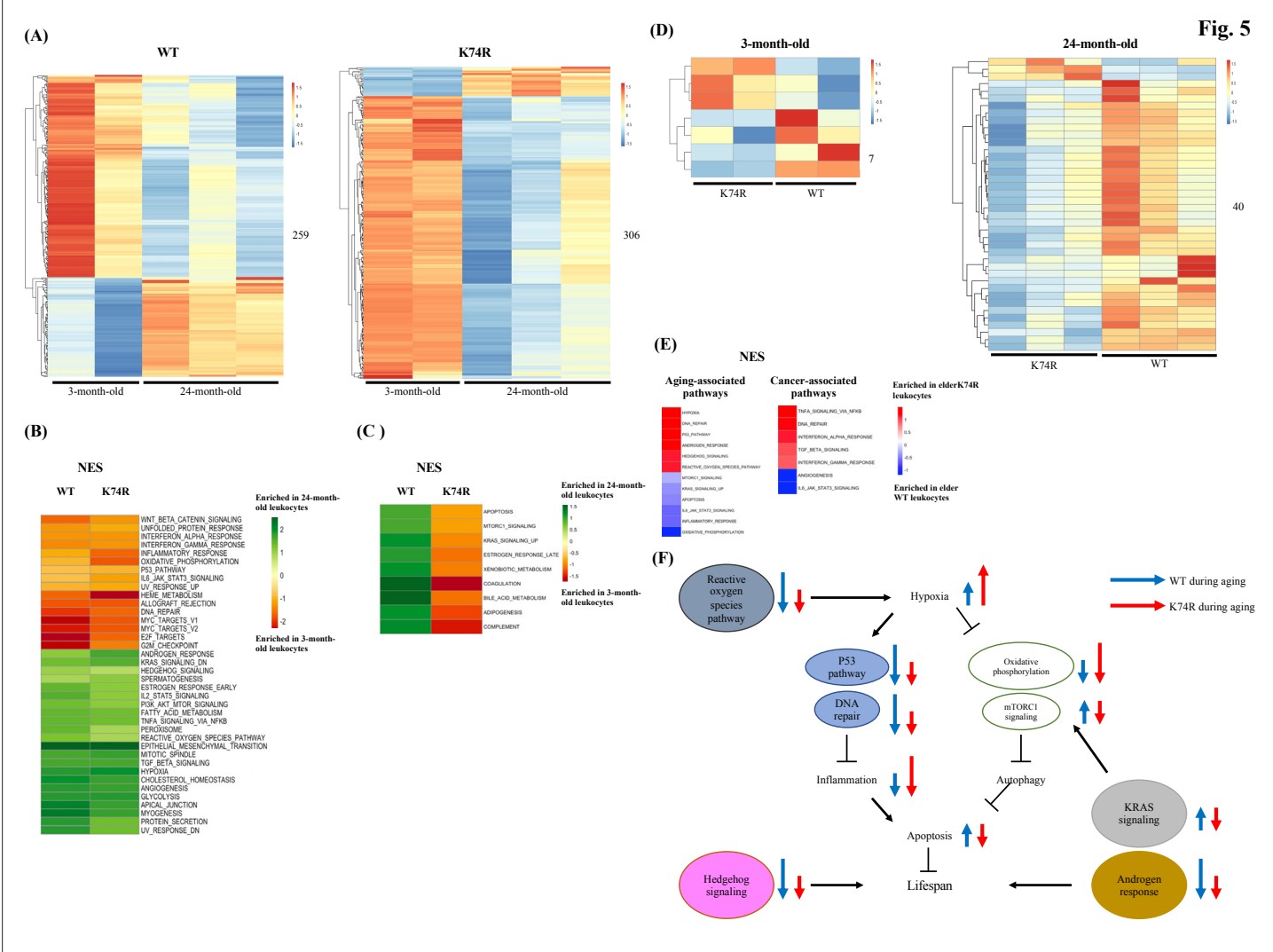

**Figure 5.** Proteomics analysis of mouse leukocytes. (**A**) Heatmap plots representing age-dependent differentially expressed proteins (DEPs) in the leukocytes from the wild-type (WT) male mice (left) and *Klf1*(K74R) male mice (right). Differential test threshold: expression fold change >1.5 and p-value <0.01. (**B, C**) Pathway analysis of the age-dependent DEPs changed in the concordant (**B**) and reverse (**C**) directions, respectively, in the WT and *Klf1*(K74R) mice. NES, normalized enrichment score. (**D**) Heatmap plots representing strain-dependent DEPs in leukocytes of 3-month-old (left) and 24-month-old (right) mice. (**E**) Pathway analyses of the strain-dependent DEPs in elder WT leukocytes (enrichment indicated by blue color) and elder *Klf1*(K74R) leukocytes (enrichment indicated by red color). The aging- and cancer-associated pathways are presented in the left and right diagrams, respectively. (**F**) A model of the regulation of the mouse lifespan by different cellular pathways in the leukocytes. The inter-relationship of 11 aging-associated cellular pathways differentially expressed in the leukocytes of *Klf1*(K74R) male mice in comparison to the WT male mice, among themselves and with respect to the regulation of lifespan (for references, see text), is depicted here. The directions and lengths of the arrows indicate changes in the individual pathways, upregulation (up arrows) or repression (down arrows), in the leukocytes during aging of the WT male mice (blue arrows) and *Klf1*(K74R) male mice (red arrows), respectively.

3-month-old mice but 40 DEPs in the 24-month-old ones. Of the 40 DEPs in the elder mice, 3 and 37 were upregulated in *Klf1*(K74R) and WT mice, respectively (*Figure 5D*). Significantly, GSEA of these DEPs showed that elder *Klf1*(K74R) leukocytes were enriched for the anti-aging pathways related to hypoxia, p53 signaling, etc. (*Yeo, 2019*; *Zhu et al., 2021*), while the elder WT leukocytes were enriched for the aging-associated pathways related to apoptosis, mTORC1 signaling, etc (*Yeo, 2019*; *Hamarsheh et al., 2020*). On the other hand, the DEPs in elder *Klf1*(K74R) leukocytes were also enriched for anti-cancer pathways related to the IFN-α response, TGF-β signaling, etc. (*Ni and Lu, 2018*; *Principe et al., 2014*; *Zitvogel et al., 2015*; *Figure 5E*), while DEPs in the elder WT leukocytes were enriched for the pro-cancer pathways related to IL-6-JAK-STAT3 signaling and angiogenesis

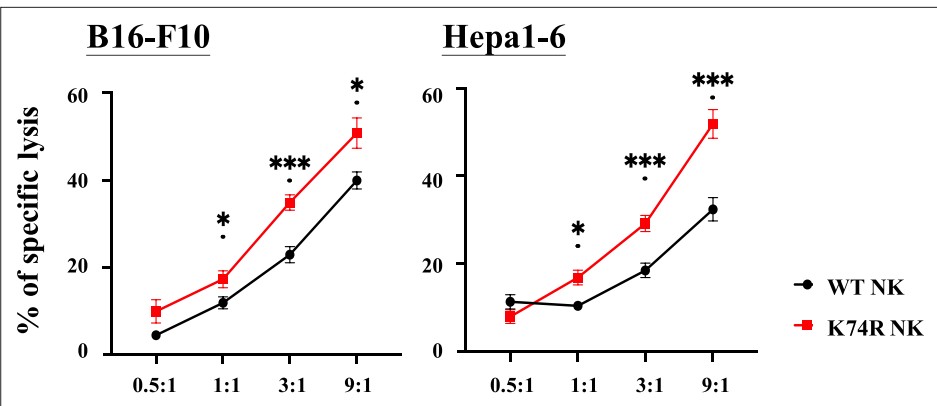

**Figure 6.** Higher in vitro cancer cell cytotoxicity of NK(K74R) cells than wild-type (WT) natural killer (NK) cells. B16-F10 and Hepa1-6 cells were labeled with Calcein-AM (BioLegend) and co-cultured with NK cells in different E:T ratio for 4 hr. Killing rates of B16-F10 and Hepa 1–6 cancer cells by NK cells were determined by the intensities of fluorescence in the medium in comparison to the controls of spontaneous release and maximal release.

(*Zhao et al., 2016*). These data together have demonstrated that *Klf1*(K74R) leukocytes contribute to their anti-cancer capability and long lifespan through several specific cellular signaling pathways.

## Higher in vitro cancer cell cytotoxicity of the NK(K74R) cells than WT NK cells

The higher anti-cancer capability of the *Klf1*(K74R) mice could result from combined contributions by different types of the hematopoietic cells, for example, the NK(K74R) cells. NK cells are effector lymphocytes of the innate immune system that can rapidly destroy tumor cells and against infectious pathogens (*Roncagalli et al., 2005*). Also, NK cells do not require pre-stimulation for the treatment of cancer cells (*Yu, 2023*). Furthermore, the NK lineage could be fully reconstituted 2 wk after BMT (*Ferreira et al., 2019*). We tested the in vitro cancer cell cytotoxicity ability of NK(K74R) cells from 3-month-old *Klf1*(K74R) mice in comparison to the WT NK cells. As shown in *Figure 6*, NK(K74R) cells exhibited significantly higher cytotoxicity against the B16-F10 melanoma cells as well as Hepa1-6 hepatocellular carcinoma cells than WT NK cells. This data, together with the higher level of NK cells in aged *Klf1*(K74R) mice (*Figure 4—figure supplement 1*), indicated that in parallel to the leukocytes (*Shyu et al., 2022*), NK cells also play an important role in the higher anti-cancer capability of the hematopoietic blood system of *Klf1*(K74R) mice, as exemplified in *Figures 2 and 3*.

## Discussion

Because of the complexity and intercrosses of the different pathways regulating the health and the aging process, genetic manipulation of non-human animals (*Bin-Jumah et al., 2022*; *Hofmann et al., 2015*), and non-genetic approaches on animals including humans (*Fontana and Partridge, 2015*; *Ocampo et al., 2016*; *Varady et al., 2022*), targeting these pathways inevitably leads to moderate-to-severe side effects such as body weight loss and adiposity. With respect to the above, the *Klf1*(K74R) mice (*Shyu et al., 2022*) are ideal as an animal model for further insightful understanding of the aging process as well as for biomedical development of new anti-aging tools and approaches. In the current study, we demonstrate that the healthy longevity of the *Klf1*(K74R) mice, as exemplified by their higher tumor resistance, is likely to be gender/genetic background independent and irrespective of the amino acid change at K74 of KLF1. Furthermore, we show that the *Klf1*(K74R) mice are resistant to carcinogenesis of not only melanoma, but also other cancers such as hepatocellular carcinoma. This correlates well with the higher cytotoxicity in vitro of NK(K74R) cells than WT NK cells against the same two types of cancer cells. Significantly, we also show that one-time transplantation of the BMMNC(BMT) could confer the recipient WT mice extended lifespan and higher tumor-resistance capability, the latter of which could be achieved by 20% of substitution of the blood of the recipient WT mice with that from the *Klf1*(K74R) mice. These findings together have interesting implications

in the development of this novel hematopoietic blood system for future medical application in anti-disease and anti-aging.

The anti-cancer capability of the *Klf1*(K74R) mice has rendered them relatively free from spontaneous cancer occurrence (*Shyu et al., 2022*), which is also reflected by their resistance to tumorigenesis of the B16-F10 cells and Hepa1-6 in the anti-cancer assays (*Figures 1–3*). Furthermore, the cancer resistance of *Klf1*(K74R) mice appears to be independent of the gender, age, and genetic background (*Figure 1*). The anti-metastasis property of the *Klf1*(K74A) mice in the pulmonary foci assay (*Figure 1D*) also indicates that the anti-cancer capability of the *Klf1*(K74R) mice is not due to the structural and/or post-translational properties of the arginine introduced at codon 74 of KLF1. Importantly, we have shown that the anti-cancer capability and the extended lifespan characteristics of *Klf1*(K74R) mice are transferrable through BMT (*Figures 2 and 3*, *Figure 2—figure supplement 1A*). In particular, we show that BMMNC from *Klf1*(K74R) mice (*Figure 2*, *Figure 2—figure supplement 1A*) could confer 2-month-old WT recipient mice with the anti-cancer capability against melanoma. Furthermore, ~20% of blood substitution would allow the recipient mice to become cancer-resistant in the pulmonary foci assay (*Figure 2D*). It is likely that homozygosity, but not heterozygosity, of K74R substitution causes one or more types of the hematopoietic cells to gain new functions, such as the higher cancer cell-killing capability of NK(K74R) cells (*Figure 6*).The data of *Figure 2D* further suggest that the amount of specific types of cells such as NK(K74R) in the 20% of blood cells carrying homozygous *Klf1*(K74R) alleles in the recipient mice upon BMT is sufficient to confer the mice a higher anti-cancer ability.

HSC therapy for different diseases (*Hifumi et al., 2017*; *Morgan et al., 2017*; *Giannaccare et al., 2020*; *Wu et al., 2020*) including cancer has been intensively explored and practiced such as leukemia and neuroblastoma (*Daver et al., 2020*; *Mora, 2022*). Also, certain characteristics of the young mice could be transferred to old mice via heterochronic parabiosis or heterochronic transplantation (*Conboy et al., 2005*; *Das et al., 2019*; *Goodell and Rando, 2015*; *Kang et al., 2020*). Similarly, plasma proteins from human umbilical cord blood can revitalize hippocampal function and neuroplasticity in aged mice (*Castellano et al., 2017*; *Mehdipour et al., 2020*). Previously, we have shown that multiple injections of LSK-HSC(K74R) from 3-month-old *Klf1*(K74R) mice to 25-month-old WT mice would extend the lifespan of the latter by about 4 months (*Shyu et al., 2014*). In the current study, we further show that 2-month-old WT mice receiving one-time transplantation of BMMNCs from 2-month-old *Klf1*(K74R) mice would live 5.5 mo longer than the controls (*Figure 2—figure supplement 1B*). Thus, transplantation/transfer of the blood MNC carrying homozygous mutation at the sumoylation site of KLF1 could be developed as a new approach for long-term anti-aging, anti-cancer, and likely rejuvenation.

The tumorigenesis resistance and long lifespan exhibited by the *Klf1*(K74R) mice are most likely due to changes in the transcription regulatory properties of the mutant KLF1(K74R) protein relative to WT (*Siatecka et al., 2007*). As exemplified in *Figure 4A and B*, expression levels of the ICGs *Pdcd* and *Cd274* in the PB, B, and T cells of *Klf1*(K74R) mice are reduced in comparison to the WT mice. Notably, cancer incidence increases with aging (*Aunan et al., 2017*), which is accompanied by increased expression of PDCD and CD274 (*Lages et al., 2010*). The lower expression of ICGs would contribute to the anti-cancer capabilities of the *Klf1*(K74R) blood to fight against cancer (*Figures 1–3*) and to extension of the lifespan of cancer-bearing mice (*Shyu et al., 2022*). Indeed, similar to ICGs, the expression levels of several cytokines in the *Klf1*(K74R) blood/serum are also different from the WT blood, and some of the changes during aging or carcinogenesis in the *Klf1*(K74R) blood are opposite to the blood/serum of WT mice (*Menees et al., 2021*; *Figure 4—figure supplement 4*).

The transcriptome data have been used to dissect the regulation of leukocyte aging (*Fabian et al., 2021*; *Aira et al., 2019*; *Schaum et al., 2020*). In addition, proteomic analysis has revealed the signaling pathways that regulate aging of specific types of leukocytes such as the lymphocyte and neutrophils cells (*Zhou et al., 2014*; *Zhang et al., 2015*). In this study, we have performed proteomics analysis of leukocytes from WT and *Klf1*(K74R) mice in two age groups and found that for the elder mice the strain-dependent DEPs in the leukocytes are enriched for a number of signaling pathways. Among these signaling pathways, at least 12 of them are closely associated with the aging process, which include hypoxia and DNA repair (*Figure 5E*). As summarized by the model in *Figure 5F*, it appears that changes in these pathways in the elder *Klf1*(K74R) leukocytes relative to the elder WT leukocytes are mostly in the direction of anti-aging. The data of *Figure 5* together strongly suggest that the KLF1(K74R) amino acid substitution causes a change in the global gene expression profile of

the leukocytes, which contributes to the high anti-cancer capability and long lifespan of the *Klf1*(K74R) mice.

In summary, we have characterized the cancer resistance of the *Klf1*(K74R) mice, among their other healthy characteristics, in relation to gender, age, and genetic background. We also have identified cell populations, gene expression profiles, and cellular signaling pathways of the WBCs of young and old mutant mice in comparison to the WT ones, which are changed in the anti-cancer and/or anti-aging directions. Finally, the transferability of the cancer resistance and extended lifespan of the mutant mice via transplantation of BMMNC suggests the possibility of future development of hematopoietic blood cells genome-edited at the conserved sumoylation site of KLF1 for anti-cancer and the extension of healthspan and/or lifespan in animals including humans.

## Animals

All the animal procedures were approved by the Institute of Animal Care and Use Committees (IACUC) at Academia Sinica. The IACUC numbers are 17-02-1052, 17-12-1142, 20-10-1528, BioTReC-110-M-005, and BioTReC-111-M-004, respectively.

# Materials and methods

## Mice

C57BL/6, B6, and FVB mice were purchased from Jackson Laboratories (Bar Harbor, ME). The B6 *Klf1*(K74R), B6 *Klf1*(K74A), and FVB *Klf1*(K74R) mice were established with the assistance of the Transgenic Core Facility, IMB, Academia Sinica, Taiwan. As described previously (*Shyu et al., 2022*), the K74R mutation was introduced by homologous recombination into exon 2 (E2) of the *Klf1* gene of B6 mice by means of a recombinant retrovirus containing the construct loxP-PGK-gb2-neo-loxP-E2 (K74R), before excising the neomycin (neo) selection marker by crossing with Ella-Cre mice. The heterozygous *Klf1*(K74R/+) mice were then crossed to obtain homozygous mutant *Klf1*(K74R/K74R) mice, hereafter termed *Klf1*(K74R) mice.

On the other hand, *Klf1*(K74A) mice were generated by using the CRISPR/Cas9 system. Female B6 mice (7–8-week-old) were mated with B6 males and the fertilized embryos were collected from the oviducts. For oligos injection, Cas mRNA (100 ng/µl), sgRNA (50 ng/µl), and donor oligos (100 ng/µl) were mixed and injected into the zygotes at the pronuclei stage. The F0 mice were genotyped by PCR and DNA sequencing. The heterozygous *Klf1*(K74A/+) mice were crossed to establish the germ-line stable homozygous *Klf1*(K74A) F1 strain.

*Klf1*(K74R) mice in the FVB background were generated using an in vitro fertilization strategy. Briefly, sperm from male B6 *Klf1*(K74R) mice was used to fertilize FVB mouse oocytes. In vitro fertilizations of FVB oocytes were carried out consecutively for five generations. The resulting chimeric mice with >90% FVB background were then crossed with FVB mice for another five generations or more.

## Cell lines

Murine B16-F10 melanoma cell line was purchased from ATCC (CRL-6475). B16-F10 cells expressing luciferase (B16-F10-luc) were generated by infection of recombinant lentivirus stably expressing luciferase luc2 under the control of hEF1A-eIF4G promoter (InvivoGen) (*Lages et al., 2010*). All cell lines were derived from cryopreserved stocks split fewer than three times and were confirmed as mycoplasma-free prior to use. B16-F10 cells were cultured at 37°C and 5% $CO_2$ in Dulbecco's Modified Eagle Medium (DMEM) supplemented with 10% fetal bovine serum (FBS), 1% penicillin/streptomycin, and 2 mM L-glutamine. B16-F10-luc cells were selected at 37°C and 5% $CO_2$ in a DMEM supplemented with 0.2 mg/ml zeocin (Invitrogen), 10% FBS, 1% penicillin/streptomycin, and 2 mM L-glutamine. The hepatocellular carcinoma cell line, Hepa1-6, was obtained from Dr. Da-Liang Ou's lab at the College of Medicine, National Taiwan University. The line was maintained at 37°C and 5% $CO_2$ in Gibico DMEM, a high glucose commercial medium containing 10% FBS (Gibico) and 1% penicillin-streptomycin.

## Pulmonary melanoma foci assay

Cultured B16-F10 melanoma cells ($1 \times 10^5$ cells/mouse) were injected into the tail vein of 8- to 9-week-old or 24-month-old *Klf1*(K74R) and WT mice with or without bone marrow transplantation

(BMT). Two weeks after injection, the mice were sacrificed and the number of tumor foci on their lungs was quantified (*Narasimhan et al., 2020*).

## Subcutaneous cancer cell inoculation assay of Hepa1-6

The protocol is similar to that used by *Han et al., 2017*. After anesthetization of the mice with 3% isoflurane, $2 \times 10^6$/50 ul of Hepa1-6 cells in serum-free DMEM were mixed with 50 ul of Corning Matrigel Matrix and injected between skin and muscle of the hind limb. The size of the tumors was measured and recorded once a week with Mitutoyo Vernier Caliper. The volume was calculated using the formula: length × weight × height × π/6.

## Flow cytometric analysis and cell sorting

Single-cell suspensions of the PB cells and spleen tissue of B6 mice were prepared by lysing red blood cells and then passing them through a 40-µm cell strainer (Falcon). BMMNCs were prepared as described in the section 'Bone marrow transplantation'. The PB cells and splenocytes were stained extracellularly for 30 min at room temperature using different combinations of the following antibodies: anti-CD45.1 (eBioscience), anti-CD45.2 (eBioscience), anti-CD3e (eBioscience), anti-CD45R (eBioscience), anti-NK1.1 (eBioscience), anti-NKp46 (eBioscience), anti-Gr-1 (eBioscience), CD11b (eBioscience), anti-PDCD (eBioscience), and anti-CD274 (eBioscience). The various hematopoietic progenitor cell compartments of bone marrow were also stained extracellularly for 30 min at room temperature by using different combinations of the following antibodies: anti-lineage (eBioscience), anti-c-Kit (eBioscience), anti-Sca-1 (eBioscience), anti-CD34 (eBioscience), and anti-Flt-3 (eBioscience). All the immunostained cells were subsequently washed with 1% phosphate buffered saline (PBS) three times and resuspended for fluorescence-activated cell sorting (FACS) analysis and sorting. Small amounts of the cell samples were run on a FACS Analyzer LSRII-12P (BD Bioscience) to determine the proportions of different cell preparations. FACS AriaII SORP (BD Bioscience) was then used to sort the indicated cell populations. The detailed gating subsets for all the cell as described above are shown in *Supplementary file 1a*. Data analysis was performed using the FlowJo software.

## RNAi knockdown of *Klf1* mRNA from T cells

CD3+ T cells isolated by sorting were cultured in RPMI 1640 medium for 1 d for recovery and then transfected with EGFP-plasmid (control), scrambled oligonucleotides (SC control), *Klf1* knockdown oligonucleotide 1 (oligo 1), or *Klf1* knockdown oligonucleotide 2 (oligo 2) in a 96-well plate for 48 hr using a LONZA electroporation kit (P3 Primary Cell 4D-Nucleofector X Kit) and machine (4D-Nucleofector Core Unit). Then, the cells were lysed using a PureLink RNA Mini kit (Life Technologies) and analyzed by RT-qPCR.

## RT-qPCR analysis

Total RNAs of B cells, T cells, and leukocytes/WBCs from the PB or spleen of *Klf1*(K74R) mice and WT mice, respectively, were extracted using a PureLink RNA Mini kit (Life Technologies). The RNAs were reverse-transcribed by means of oligo-dT primers, Maxima H Minus Reverse Transcriptase (Thermo Scientific), and SYBR Green reagents (Applied Biosystems). RT-qPCR was performed using a LightCycler Nano machine (Roche). Gene-specific primers for *Klf1*, *Pdcd*, *Cd274*, and *Gapdh* were designed using Vector NTI Advance 9 software according to respective mRNA sequences in the NCBI database (primer sequences are available upon request). The PCR primers for analysis of *Klf1* mRNA had been used and verified in a previous study (*Hung et al., 2020*). Expression levels of mRNAs were normalized to that of endogenous *Gapdh* mRNA.

## ChIP-qPCR

The ChIP-qPCR analysis followed the procedures by *Daftari et al., 1999*. Extracts were prepared from formaldehyde cross-linked mouse CD3+ T cells and erythroleukemia cell line (MEL), sonicated, and then immune-precipitated (IP) with anti-KLF1 (Thermo) or purified rabbit IgG. The chromatin DNAs in the precipitates were purified and analyzed by qPCR in the Roche LightCycle real-time system. Sequences of the DNA primers used for qPCR are shown in *Supplementary file 1*b.

## Western blotting

We adopted a previously described WB procedure (*Green and Sambrook, 2012*). Leukocytes/WBCs of *Klf1*(K74R) mice and WT B6 mice, respectively, were collected from RBC lysis buffer-treated PB. CD3+ T cells, B220+ B cells, and NK cells from the spleen were sorted using FACS AriaII SORP (BD Bioscience). The cell pellets were lysed in sample buffer and run on SDS-PAGE gels. WB with anti-KLF1 (Abcam) and anti-actin (Sigma) antibodies was then used to analyze and compare the levels of KLF1 and actin proteins.

## In vivo bioluminescence imaging

*Klf1*(K74R) and WT B6 mice were physically restrained and $1 \times 10^5$ B16-F10-luc cells/mouse were intravenously injected into their tail vein. Ten days after melanoma cell inoculation, mice were anesthetized for 5 min and injected intraperitoneally with D-luciferin (300 mg/kg of body weight). Fifteen minutes after maximum luciferin uptake, the mice were subjected to imaging of the lung and liver regions in an IVIS 50 Bioluminescence imager (Caliper Life Sciences) to determine metastatic burden. The same mice were used the next day as recipients of BMT from donor WT or *Klf1*(K74R) mice. Following BMT, bioluminescence imaging was performed on days 0, 10, 17 and 24.

## Bone marrow transplantation

BMT followed the standard protocol described in *Imado et al., 2004*. B6, CD45.1, or CD45.2 donor mice were sacrificed and their femurs were removed. Bone marrow cells were harvested by flashing the femurs with RPM I1640 medium (Gibco) using a 27-gauge needle and syringe. The cells were then incubated at 37°C for 30 min in murine complement buffer containing antibodies against B cells, T cells, and NK cells, washed twice with PBS, and then subjected to Ficoll-Paque PLUS gradient centrifugation to collect BMMNCs. BMMNCs ($1 \times 10^6$ cells/mouse) from donor mice were injected into the tail veins of recipient B6, CD45.2, or CD45.1 mice that had been exposed to total body γ-irradiation of 10, 5, or 2.5 Gy.

## Calcein-AM cytotoxicity assay of NK cells

$1 \times 10^6$ target cells (B16-F10 and Hepa1-6) were suspended in 1 ml DPBS containing 5 μM Calcein-AM (BioLegend) and incubated at 37°Cfor 30 min in the dark. The target cells were then washed by RPMI for three times and seeded at $0.3 \times 10^5$ cells per well on V-bottom 96-well plate. NK cells from the spleen were sorted by the surface markers, NK1.1+,CD3-, B220-, and NKp46+(*Supplementary file 1a*), centrifuged at $500 \times g$ for 5 min, and cultured with IL-15 in RPMI. They were co-cultured with the target cells through different E (Effector):T (Target) ratios (0.5:1, 1:1, 3:1, and 9:1). The plates were centrifuged at $250 \times g$ for 1 min to accelerate the contact of NK cells and target cells and then co-cultured for 4 hr at 37°C. Spontaneous release of Calcein-AM was measured in the target cells without NK cells, and the maximal release was determined by complete lysis of the target cells in RPMI containing 2% Triton-X 100. After co-culturing, the plates were centrifuged at $120 \times g$ for 1 min and 100 μl/well of the supernatant was transferred to a 96-well assay plate (Corning). The 488/520 nm values were determined with EnSpire (PerkinElmer).

## Bead-based multiplex assay of serum cytokines

Serum samples were obtained via submandibular blood collection and allowed to clot in uncoated tubes for 2 hr at room temperature. The tubes were centrifuged at 6000 rpm and the supernatants were collected for cytokine analysis by bead-based multiplex assay (MILLIPLEX MAP Mouse High Sensitivity T Cell Panel, Millipore) following the manufacturer's protocol (*Aira et al., 2019*).

## Protein extraction

The cell pellets were resuspended in protein extraction buffer (20 mM HEPES, 0.2% SDS, 1 mM EDTA, 1 mM glycerophosphate, 1 mM $Na_3VO_4$, and 2.5 mM $Na_4P_2O_7$) with protease inhibitor cocktail (Sigma-Aldrich) and 1 mM phenylmethylsulfonyl fluoride (PMSF). The lysates were further homogenized using a Bioruptor (Diagenode) at 4°C for 15 min, and then centrifuged at $14,000 \times g$ at 4°C for 20 min. The supernatant was transferred to a new tube before determining protein concentration by means of BCA protein assay (Pierce, Thermo Fisher). Protein aliquots were stored at –30°C until use.

## In-solution digestion

Protein solutions were first diluted with 50 mM ammonium bicarbonate (ABC) and reduced with 5 mM dithiothreitol (DTT, Merck) at 60°C for 45 min, followed by cysteine-blocking with 10 mM iodoacetamide (IAM, Sigma) at 25°C for 30 min. The samples were then diluted with 25 mM ABC and digested with sequencing-grade modified porcine trypsin (Promega) at 37°C for 16 hr. The peptides were desalted using a homemade C18 microcolumn (SOURCE 15RPC, GE Healthcare) and stored at –30°C until use.

## LC-MS/MS analysis

The desalted peptides were diluted in HPLC buffer A (0.1% formic acid in 30% acetonitrile) and loaded onto a homemade SCX column (0.6 × 5 mm, Luna 5 µm SCX 100 Å, Phenomenex). The eluted peptides were then trapped in a reverse-phase column (Zorbax 300 SB-C18, 0.3 × 5 mm; Agilent Technologies), and separated on a homemade column (HydroRP 2.5 µm, 75 µm I.D. × 15 cm with a 15 µm tip) using a multi-step gradient of HPLC buffer B (99.9% acetonitrile/0.1% formic acid) for 90 min with a flow rate of 0.3 µl/min. The LC apparatus was coupled to a 2D linear ion trap mass spectrometer (Orbitrap Elite ETD; Thermo Fisher) operated using Xcalibur 2.2 software (Thermo Fisher). Full-scan MS was performed in the Orbitrap over a range of 400–2000 Da and a resolution of 120,000 at m/z 400. Internal calibration was performed using the ion signal of $[Si(CH_3)_2O]6H^+$ at m/z 536.165365 as lock mass. The 20 data-dependent MS/MS scan events were followed by one MS scan for the 20 most abundant precursor ions in the preview MS scan. The m/z values selected for MS/MS were dynamically excluded for 40 s with a relative mass window of 10 ppm. The electrospray voltage was set to 2.0 kV, and the temperature of the capillary was set to 200°C. MS and MS/MS automatic gain control was set to 1000 ms (full scan) and 200 ms (MS/MS), or to 3 × 106 ions (full scan) and 3,000 ions (MS/MS), for maximum accumulated time or ions, respectively.

## Protein identification

Data analysis was carried out using Proteome Discoverer software (version 1.4, Thermo Fisher Scientific). The MS/MS spectra were searched against the SwissProt database using the Mascot search engine (Matrix Science, version 2.5). For peptide identification, 10 ppm mass tolerance was permitted for intact peptide masses, and 0.5 Da for CID fragment ions with an allowance for two missed cleavages arising from trypsin digestion, oxidized methionine and acetyl (protein N-terminal) as variable modifications, and carbamidomethyl (cysteine) as a static modification. Peptide spectrum matches (PSM) were then filtered based on high confidence and a Mascot search engine ranking of 1 for peptide identification to ensure an overall false discovery rate < 0.01. Proteins with single peptide hits were removed from further analysis.

## Gene set enrichment analysis

The absolute abundance of each peptide was calculated from the respective peak intensity based on the PSM abundance. The protein abundance of each sample was calculated from the sum of the peptide abundance. The abundance data were then background-corrected and normalized according to variance stabilizing transformation by using the function 'normalize_vsn' in the R package *DEP* (*Zhang et al., 2018*). Differential expression across groups was determined using the function 'test_diff' based on protein-wise linear models combined with empirical Bayes statistics. Significant DEPs were determined according to a p-value threshold of 0.01 and a fold change > 1.5. To establish functional pathways enriched across groups, normalized data for each pair of compared groups were used to perform GSEA (version 4.2.0) (*Subramanian et al., 2005*) on selected MSigDB gene sets, including Hallmark (H), curated (C2), and immunologic signature (C7) gene sets, by using the default parameters. Normalized enrichment scores (NES) were used to plot a heatmap in the R package *pheatmap* (version 1.0.12).

## Statistical analysis

Data are shown as mean ± SD or standard error of the mean (SEM). Comparisons of data under different experimental conditions were carried out using GraphPad Prism 6.0 software (GraphPad). Each error bar represents SEM unless otherwise indicated. Significant differences in tumor growth

in mouse lungs were assessed using Student's *t*-test. A difference between groups was considered statistically significant when the p-value was <0.05.

## Acknowledgements

We thank Ya-Min Lin and her colleagues at the FACS Core for their efforts in cell sorting. We also thank Taiwan Mouse Clinic facility for their expert help in the IVIS experiments. The effort by Dr. Ching-Yen Tsai at the Transgenic Core Facility, IMB, is greatly appreciated. The Immune Monitoring Core of Taipei Medical University provided great help in the analysis of the blood cell populations. We also thank the Bioinformatics Core Facility, IMB, for its efforts on the proteomics analysis, and Dr. Da-Liang Ou at the National Taiwan University for providing us with the Hepa1-6 cell line. This work was supported by Taipei Medical University (to CKJS), Academia Sinica (to CKJS), Chang-Gung Memorial Hospital Research Project (CMRPG2L0081 and CMRPG2G0311to YCS), and grants from the National Science and Technology Council, Taiwan, ROC (NSC 102-2320-B-001-010 to NSL, MOST 103-2311-B-010-003 YCS, and MOST 110-2628-B-001-024 to LYC).

## Additional information

### Competing interests

Tung-Liang Lee: is an employee of Pro-Clintech Co. Ltd. The other authors declare that no competing interests exist.

### Funding

| Funder | Grant reference number | Author |
|---|---|---|
| Taipei Medical University | | C-K James Shen |
| Academia Sinica | | C-K James Shen |
| Chang Gung Memorial Hospital | CMRPG2L0081 | Yu-Chiau Shyu |
| Chang Gung Memorial Hospital | CMRPG2G0311 | Yu-Chiau Shyu |
| National Science and Technology Council | NSC 102-2320-B-001-010 | Nan-Shih Liao |
| National Science and Technology Council | MOST 103-2311-B-010-003 | Yu-Chiau Shyu |
| National Science and Technology Council | MOST 110-2628-B-001-024 | Liuh-Yow Chen |

The funders had no role in study design, data collection and interpretation, or the decision to submit the work for publication.

### Author contributions

Jing-Ping Wang, Zheng-Sheng Lai, Data curation, Formal analysis; Chun-Hao Hung, Investigation, Writing – original draft; Yae-Huei Liou, Methodology; Ching-Chen Liu, Kun-Hai Yeh, Software, Formal analysis; Keh-Yang Wang, Investigation, Writing – review and editing; Biswanath Chatterjee, Data curation; Tzu-Chi Hsu, Data curation, Software; Tung-Liang Lee, Conceptualization; Yu-Chiau Shyu, Supervision, Writing – review and editing; Pei-Wen Hsiao, Resources, Data curation, Formal analysis; Liuh-Yow Chen, Conceptualization, Data curation; Trees-Juen Chuang, Conceptualization, Software, Formal analysis, Writing – review and editing; Chen-Hsin Albert Yu, Software, Formal analysis, Supervision; Nan-Shih Liao, Conceptualization, Resources, Methodology; C-K James Shen, Conceptualization, Supervision, Project administration, Writing – review and editing

### Author ORCIDs

Chun-Hao Hung ⓘ https://orcid.org/0000-0003-0407-8828
Yu-Chiau Shyu ⓘ https://orcid.org/0000-0003-1747-2226

C-K James Shen [ID] https://orcid.org/0000-0001-6515-5837

### Ethics

All the animal procedures were approved by the Institute of Animal Care and Use Committees, IACUC, at Academia Sinica. The IACUC numbers are 17-02-1052, 17-12-1142, 20-10-1528, BioTReC-110-M-005, and BioTReC-111-M-004, respectively.

Reviewer #1 (Public Review): https://doi.org/10.7554/eLife.88275.3.sa1
Reviewer #2 (Public Review): https://doi.org/10.7554/eLife.88275.3.sa2
Reviewer #3 (Public Review): https://doi.org/10.7554/eLife.88275.3.sa3
Author response https://doi.org/10.7554/eLife.88275.3.sa4

## Additional files

### Supplementary files

• Supplementary file 1. Cell surface markers of different hematopoietic blood cells. (a) Cell surface markers of different hematopoietic blood cells. (b) DNA primers used in ChIP-qPCR experiments.

• MDAR checklist

### Data availability

All data that support the findings of this study are available from Dryad https://doi.org/10.5061/dryad.mkkwh717j.

The following dataset was generated:

| Author(s) | Year | Dataset title | Dataset URL | Database and Identifier |
|---|---|---|---|---|
| Wang JP, Hung CH, Liou YH, Liu CC, Yeh KH, Wang KY, Lai ZS, Chatterjee B, Hsu TC, Lee TL, Shyu YC, Hsiao PW, Chen LY, Shen CKJ, Chuang TJ, CHA Yu, Liao NS | 2024 | Long-term Hematopoietic Transfer of the Anti-Cancer and Lifespan-Extending Capabilities of A Genetically Engineered Blood System by Transplantation of Bone Marrow Mononuclear Cells | https://doi.org/10.5061/dryad.mkkwh717j | Dryad Digital Repository, 10.5061/dryad.mkkwh717j |

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
