## [Editor Report · eLife assessment]

This **useful** article focuses on understanding how an Eklf mutation confers anti-cancer and longevity advantages in vivo. The data demonstrate that Eklf (K74R) imparts such advantages in a background- and age-independent manner in both female and male mice, and that the benefits are transferable by bone marrow transplantation. Despite added data since a previous version, the article unfortunately remains **incomplete** as it is still unclear whether Eklf affects resistance to malignant progression/metastasis by modulating Pd1 or Pdl1 or by increasing NK cells. The authors provide evidence that supports in principle both mechanisms, and they do not resolve which mechanism is primarily involved.

---

## [Referee Report · Reviewer #1 (Public Review)]

The authors Wang et al. present a study of a mouse model K74R that they claim can extend the life span of mice, and also has some anti-cancer properties in some standrad models of melanoma and hepatocellular carcinoma. Importantly, this mechanism seems to be mediated by the hematopoietic system, and protective effects can be transferred with bone marrow transplantation.

The authors have now adapted their manuscript reflecting the novelties of these studies. Overall, the study is a continuation and also corroboration of previous work, without clinical data yet. The authors have now expanded their work to a second mouse model, which strengthens their data.

---

## [Referee Report · Reviewer #2 (Public Review)]

The manuscript by Wang et al., follows up on the group's previous publication on KLF1 (EKLF) K47R mice and reduced susceptibility to tumorigenesis and increased life span (Shyu et al., Adv Sci (Weinh). Sep 2022;9(25):e2201409. doi:10.1002/advs.202201409). In the current manuscript, the authors have described these phenotypes in the context of age, gender, genetic background, and hematopoietic transplantation of bone marrow mononuclear cells. Despite the revisions, significant conceptual concerns still remain in the study that make the inferences in the manuscript less convincing. Major concerns are listed below.

Major concerns:

1. The authors mention more than once in the manuscript that KLF1 is expressed in range of blood cells including hematopoietic stem cells, megakaryocytes, T cells and NK cells. In the case of megakaryocytes, studies from multiple labs have shown that while EKLF is expressed megakaryocyte-erythroid progenitors, EKLF is important for the bipotential lineage decision of these progenitors, and its high expression promotes erythropoiesis, while its expression is antagonized during megakaryopoiesis. In the case of HSCs, the authors reference to their previous publication for KLF1's expression in these cells- however, in this study nor in the current study, there is no western blot documented to convincingly show that KLF1 protein is expressed at detectable levels in these cells. For T cells, the authors have referenced a study which is based on ectopic expression of KLF1. For NK cells, the authors reference bioGPS: however, upon inspection, this is also questionable. As part of the revision, the authors have provided western blots in supplemental figure S4. However, these blots are difficult to interpret, since the EKLF bands for NK cells, and T cells are very faint and since the positive control EKLF band from MEL erythroid cell lysates is oversaturated, to interpret the data clearly. Therefore, although a quantification is shown, the representative blot included for EKLF protein levels is not convincing.

2. The current study rests on the premise that KLF1 is expressed in HSCs, NK cells and leukocytes, and the references cited are not sufficient to make this assumption, for the reasons mentioned in the first point. Therefore, the authors were asked to show both KLF1 mRNA and protein levels in these cells, and also compare them to the expression levels seen in KLF1 wild type erythroid cells along with knockout erythroid cells as controls, for context and specificity. The authors have now included western blots and mRNA levels and have compared it to MEL erythroid cells. This data raises additional questions. Overall, the mRNA levels in CD3+ T cells and B220+ B cells are approximately 3000 fold lower than MEL erythroid cells. Based on the information provided, although unclear, the assumption is that the MEL cell extracts are from undifferentiated cells. Therefore, this raises questions on the inference that the healthy aging phenotype is a result of cell intrinsic effects, since EKLF expression in these cells of interest is extremely low. This also allows for the consideration for potential systemic/indirect effects.

3. In the discussion, the authors make broad inferences that go beyond the data shown in the manuscript. For example, they mention that the tumorigenesis resistance and long lifespan is most likely due to changes in transcription regulatory properties and changes in global gene expression profile of the mutant protein relative to WT leukocytes. And based on reduced mRNA levels of Pd-1 Pd-l1 genes in the CD3+ T cells and B220+ B cells from mutant mice, they "assert" that EKLF is an upstream regulator of these genes and regulates the transcriptomes of a diverse range of hematopoietic cells. The authors were asked to perform a ChIP assay to show whether WT EKLF binds on these genes in these cells, and whether this binding is reduced or abolished in the mutant cells, to substantiate the above statements. The authors have now included a ChIP assay in Figure S5. The data on WT EKLF and K74R EKLF on Pd-1 promoter shows that both forms of EKLF bind at similar levels. Therefore, the mechanism remains unclear, and there is insufficient discussion on how their data support cell intrinsic differences in transcriptional regulation between WT and mutant EKLF.

---

## [Referee Report · Reviewer #3 (Public Review)]

Hung et al provide a well-written manuscript focused on understanding how Eklf mutation confers anticancer and longevity advantages in vivo.

The authors were responsive to the reviewers comments in some aspects. However, the manuscript continues to suffer from significantly overstated claims that are not mitigated in the revision. While additional data has been added, it is unclear how this new data provides clarity to the overall premise of this observational study. Importantly, the authors have added a second model of hepatocellular carcinoma with findings that are consistent with the melanoma model previously reported. In addition, they make more clear that the previously published manuscript on this subject was use of older donors for BMT while now they use younger donors. This is at best incremental. It remains unclear whether Eklf exerts its effect on resistance to malignant progression / metastasis by modulating Pd1 or Pdl1 vs. increasing NK cells as the authors provide evidence of both and do not resolve which mechanism is primarily involved. Finally, there is no evidence that Eklf mutation confers "an anti-disease and anti-aging" effect as at best the data provides evidence of resistance to malignant progression / metastasis in melanoma and hepatoma models.

The work is impactful as it provides evidence of anticancer effect of a specific hematological mutation but the mechanism by which this occurs is not completely elucidated by this work.

---

## [Author Response]

The following is the authors’ response to the original reviews.

We thank the reviewers and the editors for their constructive and critical comments/ suggestions regarding our paper. We have since extensively revised the manuscript accordingly, including the addition of new experimental data. Hope the readers, reviewers, and editors are now satisfied with the quality and significance of the revised paper.

Our responses to the eLife assessment and the reviewers’ comment as well as the details of the revisions are described below.

Wang et al present a useful manuscript that builds modestly on the group's previous publication on KLF1 (EKLF) K47R mice focused on understanding how Eklf mutation confers anticancer and longevity advantages in vivo (Shyu et al., Adv Sci (Weinh). 2022). The data demonstrates that Eklf (K74R) imparts these advantages in a background, age, and gender independent manner, not the consequence of the specific amino acid substitution, and transferable by BMT. However, the authors overstate the meaning of these results and the strength of evidence is incomplete, since only a melanoma model of cancer is used, it is unclear why only homozygous mutation is needed when only a small fraction of cells during BMT confer benefit, they do not show EKLF expression in any cells analyzed, and the PD-1 and PDL-1 experiments are not conclusive. The definitive mechanism relative to the prior publication from this group on this topic remains unclear.

The issues in the assessment by the editor on our paper were also brought up by the reviewers. We have taken care of them by carrying out new experiments as well as rewriting of the paper to highlight the rationales and novel aspects of the current study, as described below in our responses to the three reviewers.

**Public Reviews:**

**Reviewer #1 (Public Review):**
The authors Wang et al. present a study of a mouse model K74R that they claim can extend the life span of mice, and also has some anti-cancer properties. Importantly, this mechanism seems to be mediated by the hematopoietic system, and protective effects can be transferred with bone marrow transplantation.The authors need to be more specific in the title and abstract as to what is actually novel in this manuscript (a single tumor model), and what relies on previously published data (lifespan). Because many of these claims derive from previously published data, and the current manuscript is an extension of previously published work. The authors need to be more specific as to the actual data they present (they only use the B16 melanoma model) and the actual novelty of this manuscript.Especially experiments on life span are published and not sufficiently addressed in this actual paper, as the title would suggest.

Indeed important to point out the novelty of this paper in comparison to the previous paper. First, we have modified the title, the abstract, and the text so to emphasize that the extended lifespan as well as tumor resistance could be transferred by from Eklf(K74R) mice to WT mice by a single transplantation of the Eklf(K74R) bone marrow mononuclear cells (BMT) to the WT mice at their young age (2 months).

We now also provide several new experimental data including the one demonstrating that Eklf(K74R) mice are resistant to tumorigenesis of hepatocellular carcinoma as well (new Fig. 1E). These points are elaborated in more details below in my responses to the reviewers’ comments/ suggestions.

**Reviewer #2 (Public Review):**
The manuscript by Wang et al. follows up on the group's previous publication on KLF1 (EKLF) K47R mice and reduced susceptibility to tumorigenesis and increased life span (Shyu et al., Adv Sci (Weinh). Sep 2022;9(25):e2201409. doi:10.1002/ advs.202201409). In the current manuscript, the authors have described the dependence of these phenotypes on age, gender, genetic background, and hematopoietic translation of bone marrow mononuclear cells. Considering the current study is centered on the phenotypes described in the previous study, the novelty is diminished. Further, there are significant conceptual concerns in the study that make the inferences in the manuscript far less convincing. Major concerns are listed below:1. The authors mention more than once in the manuscript that KLF1 is expressed in range of blood cells including hematopoietic stem cells, megakaryocytes, T cells and NK cells. In the case of megakaryocytes, studies from multiple labs have shown that while EKLF is expressed megakaryocyte-erythroid progenitors, EKLF is important for the bipotential lineage decision of these progenitors, and its high expression promotes erythropoiesis, while its expression is antagonized during megakaryopoiesis. In the case of HSCs, the authors reference to their previous publication for KLF1's expression in these cells- however, in this study nor in the current study, there is no western blot documented to convincingly show that KLF1 protein is expressed at detectable levels in these cells. For T cells, the authors have referenced a study which is based on ectopic expression of KLF1. For NK cells, the authors reference bioGPS: however, upon inspection, this is also questionable.1. The current study rests on the premise that KLF1 is expressed in HSCs, NK cells and leukocytes, and the references cited are not sufficient to make this assumption, for the reasons mentioned in the first point. Therefore, the authors will have to show both KLF1 mRNA and protein levels in these cells, and also compare them to the expression levels seen in KLF1 wild type erythroid cells along with knockout erythroid cells as controls, for context and specificity.

Regarding the novelties of the current story. Besides demonstration of the independence of the healthy longevity characteristics on age, gender, and genetic background, as exemplified by the tumor resistance, another novelty of the current study is that the healthy longevity characteristics, in particular the tumor resistance and extended lifespan, could be transferred by one-time long-term transplantation of the Eklf(K74R) bone marrow mononuclear cells from young Eklf(K74R) mice to young WT mice. Also, since submission of the last version of the paper, we have carried out new experiments, including the characterization of the anti-cancer capability of NK cells (new Fig. 6) as well as assay of the tumor-resistance of Eklf(K74R) mice to hepatocellular carcinoma (new Fig. 1E), etc.

We have also modified the title, Abstract, and different parts of the text to highlight the novelties of the current study.

As to the expression of EKLF in different hematopoietic blood cell types, we have now added a paragraph in Result (p.6 and p.7) describing what have been known in literature in relation to our data presented in the paper. Importantly, following the reviewer’s comments, we have since carried out Western blot analysis of EKLF expression in NK, T, and B cells (p. 6, p.7 and new Fig. S4B). Also noted is that the level of EKLF in B cells is very low and only could be detected by RT-qPCR (Fig. S4C) and RNA-Seq (Bio-GPS database)

1. To get to the mechanism driving the reduced susceptibility to tumorigenesis and increased life span phenotypes in EKLF K74R mice, the authors report some observations- However, how these observations are connected to the phenotypes is unclear.a. For example, in Figure S3, they report that the frequency of NK1.1+ cells is higher in the mutant mice. The significance of this in relation to EKLF expression in these cells and the tumorigenesis and life span related phenotypes are not described. Again, as mentioned in the second point, KLF1 protein levels are not shown in these cells.b. In Figure 4, the authors show mRNA levels of immune check point genes, PD-1 and PD-l1 are lower in EKLF K74R mice in PB, CD3+ T cells and B220+ B cells. Again, the questions remain on how these genes are regulated by EKLF, and whether and at what levels EKLF protein is expressed in T cells and B cells relative to erythroid cells. Further, while the study they reference for EKLF's role in T cells is based on ectopic expression of EKLF in CD4+ T cells, in the current study, CD3+ T cells are used. Also, there are no references for the status of EKLF in B cells. These details are not discussed in the manuscript.

Regarding this part of the questions and comments by the reviewer.

First, we have since assayed the effect of the K74R substitution of EKLF on the in vitro cancer cell-killing ability of NK cells (termed NK1.1 cells in the previous version). The data showed that NK(K74R) cells have higher ability than the WT NK cells (new Fig. 6). This property together with the higher expression level of NK(K74R) cells in 24 month-old Eklf (K74R) mice than NK cells in 24 month-old WT mice would contribute to the higher tumor-resistance of the Eklf (K74R) mice. This point is also addressed on p. 8 andp.9.

Second, as stated in previous sections, we have since carried out comparative Western blot analysis of the expression of EKLF protein in NK, CD3 T, and B cells of the WT and Eklf(K74R) mice, respectively (please see the new Fig. S4B). Also, description regarding what are known in literature in relation to our data on the expression of EKLF protein/ Eklf mRNA in different types of hematopoietic blood cells is now included in the Result (please see p.6 and p.7). Notably though, the level of EKLF protein in B cells was too low to be detected by WB (Fig. S4B).

1. The authors perform comparative proteomics in the leukocytes of EKLF K74R and WT mice as shown in Figure S5. What is the status of EKLF levels in the mutant lysate vs wild type lysates based on this analysis? More clarity needs to be provided on what cells were used for this analysis and how they were isolated since leukocytes is a very broad term.

The leukocytes used by us were isolated from the peripheral blood after removal of red blood cells, as described in the Materials and Methods.

Also, the Western blot analysis of EKLF expression in the lysates of leukocytes/ white blood cells (WBC) has been shown previously, now presented in the new Figure S4A.

1. In the discussion the authors make broad inferences that go beyond the data shown in the manuscript. They mention that the tumorigenesis resistance and long lifespan is most likely due to changes in transcription regulatory properties and changes in global gene expression profile of the mutant protein relative to WT leukocytes. And based on reduced mRNA levels of Pd-1 Pd-l1 genes in the CD3+ T cells and B220+ B cells from mutant mice, they "assert" that EKLF is an upstream regulator of these genes and regulates the transcriptomes of a diverse range of hematopoietic cells. The lack of a ChIP assay to show binding of WT EKLF on genes in these cells and whether this binding is reduced or abolished in the mutant cells, make the above statements unsubstantiated.

We have since carried out ChIP-PCR analysis of EKLF-binding in the Pd-1 promoter (new Fig. S5). The data showed that EKLF was bound on the CACCC box at -103 of the promoter in WT CD3+T as well as in CD3+T(K74R) cells. This result is discussed on p.7.

1. Where westerns are shown, the authors need to show the molecular weight ladder, and where qPCR data are shown for EKLF, it will be helpful to show the absolute levels and compare these levels to those in erythroid cells, along the corresponding EKLF knock out cells as controls.

We have since included the molecular weight markers by the side of Western blots in Fig. S4. Also, we have added a new figure (Fig.S4C) showing the comparison of the expression levels of Eklf mRNA in B cells and CD3+ T cells to the mouse erythroleukemia (MEL) cells, as analyzed by RT-qPCR.

Also, as indicated now in the Material and Methods section, the specificity of the primers used for RT-qPCR quantitation of mouse Eklf mRNA has been validated before by comparative analysis of wild type and EKLF-knockout mouse erythroid cells (Hung et al., IJMS, 2020).

1. Figure S1D does not have a figure legend. Therefore, it is unclear what the blot in this figure is showing. In the text of the manuscript where they reference this figure, they mention that the levels of the mutant EKLF vs WT EKLF does not change in peripheral blood, while in the figure they have labeled WBCs for the blot, and the mRNA levels shown do seem to decrease in the mutant compared to WT peripheral blood.

We apologize for this ignorance on our side. The data shown in the original Fig. SID (new Fig. S4A) are from Western blot analysis of EKLF protein and RT-qPCR analysis of Eklf mRNA in leukocytes/ white blood cells (WBC) isolated from the peripheral blood samples. We have now added back the figure legend and also rewritten the corresponding description in the text on p.6.

**Reviewer #3 (Public Review):**
Hung et al provide a well-written manuscript focused on understanding how Eklf mutation confers anticancer and longevity advantages in vivo. The work is fundamental and the data is convincing although several details remain incompletely elucidated. The major strengths of the manuscript include the clarity of the effect and the appropriate controls. For instance, the authors query whether Eklf (K74R) imparts these advantages in a background, age, and gender dependent manner, demonstrating that the findings are independent. In addition, the authors demonstrate that the effect is not the consequence of the specific amino acid substitution, with a similar effect on anticancer activity. Furthermore, the authors provide some evidence that PD-1 and PDL-1 are altered in Eklf (K74R) mice.

Here we thank the encouraging comments by this reviewer.

Finally, they demonstrate that the effects are transferrable with BMT. Several weaknesses are also evidence. For instance, only melanoma is tested as a model of cancer such that a broad claim of "anti-cancer activity" may be somewhat of an overreach.

We have now included new data showing that the Eklf(K74R) mice also carry a higher anti-cancer ability against hepatocellular carcinoma than the WT mice (new Fig. 1E).

It is also unclear why a homozygous mutation is needed when only a small fraction of cells during BMT can confer benefit. It is also difficult to explain how transplanted donor Eklf (K74R) HSCs confer anti-melanoma effect 7 and 14 days after BMT.

First, these two observations not necessarily conflict with each other. It is likely that homozygosity, but not heterozygosity, of the K74R substitution in EKLF allows one or more types of hematopoietic blood cells to gain new functions, e.g. the higher cancer cell- killing capability of NK(K74R) cells (new Fig. 6), that help the mice to live long and healthy. Also, the data in Fig. 2D indicated that as low as 20% of the blood cells carrying homozygous Eklf(K74R) alleles in the recipient mice upon BMT could be sufficient to confer the mice a higher anti-cancer capability, likely in part due to cells such as NK(K74R). These points are now clarified in Discussion (p.9 and p.10).

Second, we think the NK(K74R) cells contributed a significant part to the anti-cancer capability of the transplanted Eklf(K74R) blood in the recipient WT mice. As documented in some literature, e.g. Ferreira et al., Journal of Molecular Medicine (2019), the hematopoietic lineage of the NK cells would be fully reconstituted as early as 2 weeks after BMT. Of course, there could be other still unknown factors/ cells that also contribute to the tumor-resistance of the recipient mice at 7 day following BMT. This point is now touched upon on p.8 and p.9.

Furthermore, it would be useful to see whether there are virulence marker alterations in the melanoma loci in WT vs Eklf (K74R) mice.

As responded in the Public Reviews, we will analyze this in future together with other types of tumors in a separate study.

Finally, the data in Fig 4c is difficult to interpret as decreased PD-1 and PDL-1 after knockdown of EKLF in vitro is not a useful experiment to corroborate how mutation without changing EKLF expression impacts immune cells. The work is impactful as it provides evidence that healthspan and lifespan may be modulated by specific hematological mutation but the mechanism by which this occurs is not completely elucidated by this work.

As described in a previous section, we have since also carried out ChIP-qPCR analysis of the binding of WT EKLF and EKLF (K74R) on the Pd-1 promoter (new Fig. S5).

**Reviewer #1 (Recommendations For The Authors):**
The authors present interesting melanoma model data but need to tone down their claim of multiple effects of their model system. It needs to be clear what is new and what is previously known.

As respond in the Public Reviews, we have since added new data on the tumor resistance of the Eklf(K74R) mice to hepatocellular carcinoma (new Fig. 1E). We have also modified the title as well as highlighted the novel points in the Abstract and text of the revised draft.

**Reviewer #2 (Recommendations For The Authors):**
In addition to the major concerns listed in the public review, the minor concerns that the authors could address are listed below:1. Will be helpful to describe why was the pulmonary melanoma focus assay chosen for metastasis assay?

We now describe on p. 4 the rationale behind the initial choice of this assay for analysis of the anti-cancer capability of the Eklf(K74R) mice. Also, we have since included data from experiment using the subcutaneous cancer cell inoculation assay for comparative analysis of the anti-hepatocellular carcinoma capability of Eklf(K74R) and WT mice (Fig. 1E and p.5).

1. Reference #61 for B16-F10-luc cells cited in the methods does not have details on the generation of these cells. What these cells are and why this model was chosen needs to be described.

Sorry about not providing this information before. We now describe the generation of B16F10-luc cells in the Material and Methods section (p.13). The rationale of choosing the B16-F10 cells for the pulmonary lung foci assay is also added on p.4.

1. The DNA binding consensus site for EKLF needs to be expanded in the introduction.

This part has been taken care of now on p.13.

**Reviewer #3 (Recommendations For The Authors):**
Hung et al provide a well-written manuscript focused on understanding how Eklf mutation confers anticancer and longevity advantages in vivo. The work is fundamental and the data is convincing although several details remain incompletely elucidated.1. Only melanoma is tested as a model of cancer such that a broad claim of "anti-cancer activity" may be somewhat of an overreach. The authors, therefore, need to provide evidence of a second type of malignancy to which Eklf mutation confers anticancer and longevity advantages or temper the claims in the discussion that the effect still needs to be tested in non-melanoma cancer models to determine the broad anti-cancer effect.

As responded in the Public Reviews, we have since shown that Eklf(K74R) mice also exhibited a higher resistance to the carcinogenesis of hepatocellular carcinoma (new Fig. 1E).

1. Why is a homozygous mutation needed when only a small fraction of cells during BMT can confer benefit of Eklf mutation? Is there evidence that the cellular effect is binary but only a few such cells are needed? This is confusing and requires further clarification.

As responded in the Public Reviews, these two observations not necessarily conflict with each other. It is likely that homozygosity, but not heterozygosity, of the K74R substitution in EKLF allows one or more types of hematopoietic blood cells to gain new functions, e.g. the higher cancer cell- killing capability of NK(K74R) cells (new Fig. 6), that help the mice to live long and healthy. Also, the data in Fig. 2D indicated that as low as 20% of the blood cells carrying homozygous Eklf(K74R) alleles in the recipient mice upon BMT could be sufficient to confer the mice a higher anti-cancer capability, likely in part due to cells such as NK(K74R). This point is now clarified in Discussion (p.9).

1. BMT typically requires at least 3-4 weeks to reconstitute the marrow compartment but the authors are able to see effects of Eklf mutation as early as 7 days following BMT. This is surprising and brings into question the mechanism of effect.

As responded in the Public Reviews, we think the NK(K74R) cells contributed a significant part to the anti-cancer capability of the transplanted Eklf(K74R) blood in the recipient WT mice. As documented in some literature, e.g. Ferreira et al., Journal of Molecular Medicine (2019), the hematopoietic lineage of the NK cells would be fully reconstituted as early as 2 weeks after BMT. Of course, there could be other still unknown factors/ cells that also contribute to the tumor-resistance of the recipient mice at 7 day following BMT (please see discussion of this point on p. 9).

1. It would be useful to see whether there are virulence marker alterations in the melanoma loci in WT vs Eklf (K74R) mice.

As responded in the Public Reviews, we will analyze this in future together with other types of tumors in a separate study.

1. The data in Fig 4c is difficult to interpret as decreased PD-1 and PDL-1 after knockdown of EKLF in vitro is not a useful experiment to corroborate how mutation WITHOUT changing EKLF expression impacts immune cells.

Indeed, the RNAi knockdown experiment only demonstrated a positive regulatory role of EKLF in Pd1/Pd-l1 gene expression. We have followed the reviewer’s suggestion and carried out ChIP-qPCR analysis and shown that the factor is bound on the Pd-1 promoter in both WT CD3+T cells and CD3+T(K74R) cells (new Fig. S5). We briefly discuss these data on p.7 in relation to the possible effect of K74R substitution of EKLF on Pd-1 expression.

We have now further clarified this point on p. 7.